# Subversion of GBP-mediated host defense by E3 ligases acquired during *Yersinia pestis* evolution

Shiyang Cao[1,3], Yang Jiao[1,3], Wei Jiang[2], Yarong Wu [1], Si Qin[1], Yifan Ren[1], Yang You[1], Yafang Tan[1], Xiao Guo[1], Hongyan Chen[1], Yuan Zhang[1], Gengshan Wu[1], Tong Wang[1], Yazhou Zhou[1], Yajun Song [1], Yujun Cui [1], Feng Shao [2], Ruifu Yang [1] ✉ & Zongmin Du [1] ✉

Plague has caused three worldwide pandemics in history, including the Black Death in medieval ages. *Yersinia pestis*, the etiological agent of plague, has evolved a powerful arsenal to disrupt host immune defenses during evolution from enteropathogenic *Y. pseudotuberculosis*. Here, we find that two functionally redundant E3 ligase of *Y. pestis*, YspE1 and YspE2, can be delivered via type III secretion injectisome into host cytosol where they ubiquitinate multiple guanylate-binding proteins (GBPs) for proteasomal degradation. However, *Y. pseudotuberculosis* has no such capability due to lacking functional YspE1/2 homologs. YspE1/2-mediated GBP degradations significantly promote the survival of *Y. pestis* in macrophages and strongly inhibit inflammasome activation. By contrast, $Gbp^{chr3-/-, chr5-/-}$ macrophages exhibit much lowered inflammasome activation independent of YspE1/2, accompanied with an enhanced replication of *Y. pestis*. Accordingly, $Gbp^{chr3-/-, chr5-/-}$ mice are more susceptible to *Y. pestis*. We demonstrate that *Y. pestis* utilizes E3 ligases to subvert GBP-mediated host defense, which appears to be newly acquired by *Y. pestis* during evolution.

Plague is a zoonotic disease that has caused three pandemics in human history including the notorious Black Death in medieval Europe[1]. The etiological agent of plague, *Yersinia pestis*, has recently evolved from the enteropathogenic *Yersinia pseudotuberculosis*[2,3], a foodborne enteropathogen causing self-limited gastrointestinal disease, to a flea borne and lethal pathogen. Although *Y. pestis* and *Y. pseudotuberculosis* cause distinct disease and adopt different typical routes of transmission, they are closely related at the genetic level since *Y. pestis* is deemed to be a highly uniform clone of *Y. pseudotuberculosis*[3,4]. The two pathogens shared a common 70-Kb plasmid encoding the type III secretion system (T3SS), an indispensable virulence mechanism of pathogenic yersiniae[5]. *Y. pestis* has acquired two additional plasmids during evolution, termed pMT1 and pPCP1. pMT1 encodes the anti-phagocytic F1 capsular antigen and the murine toxin gene (*ymt*) that is involved in survival and replication of *Y. pestis* inside the flea[6]. pPCP1

encodes the plasminogen activator (Pla protease) that is required for the dissemination of *Y. pestis* from subcutaneous infection sites as well as the primary pneumonic plague[7–9]. Acquisitions of pMT1 and pPCP1 provide the prerequisite for *Y. pestis* to evolve the capability to establish a systemic infection via flea-bite transmission from a local lymphadenitis[10–13]. Meanwhile, *Y. pestis* has underwent a significant genome reductive evolution by pseudogenization of several virulence factors essential for gastrointestinal colonization and evolving novel virulence mechanisms to arm itself[10,14]. For instance, Inv and YadA are adhesins indispensable for enteropathogenic yersiniae to colonize the Peyer's patches and mesenteric lymph nodes; however, both *inv* and *yadA* are inactivated in *Y. pestis*[15,16]. Due to the loss of *lpxL* gene, Lipid A of *Y. pestis* lipopolysaccharide (LPS) switches from a hexa- to tetra-acylated form that is poorly recognized by Toll-like receptor 4 (TLR4) at the natural mammalian host temperature (37 °C)[17,18].

---

[1]State Key Laboratory of Pathogen and Biosecurity, Beijing Institute of Microbiology and Epidemiology, 100071 Beijing, China. [2]National Institute of Biological Sciences, Beijing 102206, China. [3]These authors contributed equally: Shiyang Cao, Yang Jiao. ✉e-mail: ruifuyang@gmail.com; zmduams@163.com

*Y. pestis* is a facultative intracellular pathogen readily engulfed by macrophages when initially invading the host, and treatment with IFN-γ increases the killing activity of host professional phagocytes to *Y. pestis* residing inside cells[19,20]. At the early stage of infection, it is critical for *Y. pestis* to survive in macrophages, whereby it evades the host immune surveillance and coordinate the gene expressions for the rapid adaption to the adverse environments within the host. It has been reported that both *Y. pestis* and *Y. pseudotuberculosis* can survive and replicate inside the host macrophages[21]. Although the transactional regulator PhoP and RipA in *pgm* locus have been implicated in survival of *Y. pestis* in macrophages[22–24], the molecular mechanisms involved are still poorly understood. Guanylate-binding proteins (GBPs) are IFN-γ induced GTPase family members that have been shown to play vital roles in cell-autonomous antimicrobial defense against bacterial, virus and protozoans pathogens[25,26]. Bacterial pathogens including *Shigella, Francisella, Legionella* and *Salmonella* have evolved multiple virulence effectors to disrupt GBP-mediated host defense[27–29]. For instance, *Shigella* IpaH9.8, a T3SS effector, can ubiquitinate hGBPs for proteasomal degradation to promote the intracellular replication and spread of bacteria[30,31]. In this study, we showed that two Novel E3 ubiquitin ligase (NEL) family members, YP_3416 and YP_3418, which were named as Yersinia secreted E3 ligase 1 (YspE1) and YspE2 in this study, target GBPs for degradation of both human GBPs (hGBPs) and murine GBPs (mGBPs), to promote the survival inside the macrophages and strongly inhibit the inflammasome activation. However, *Y. pseudotuberculosis*, the closely related ancestor of *Y. pestis* cannot disrupt the GBP mediated cell-autonomous defense due to the lack of activity to degrade GBPs. Our data demonstrate the importance of GBPs for innate host defenses against plague. More importantly, we further reveal that acquisition of functionally redundant E3 ligase, YspE1 and YspE2, that can degrade multiples GBPs is one of the critical steps for virulence evolution of *Y. pestis* since this capability appears to be gradually stabilized after divergence from *Y. pseudotuberculosis*.

## Results

### YspE1 and YspE2 are delivered by T3SS into host cytosol where they target hGBP1 for proteasomal degradation

Our previous results showed that deletion of *yspE1/yspE2* genes attenuated the virulence of *Y. pestis*[32], but the underlying mechanism is

unresolved. We also revealed that YspE1 and YspE2 belong to the NEL family and they show strong E3 ligase activity dependent on a conserved Cys residue of CEDR motif of NEL domain[32–34], while YP_3417 only harbors the leucine-rich repeat (LRR) domain that is known to be responsible for the substrates recognition. A T3SS mutant of *Y. pestis* defective in Yops secretion also secreted much less YspE1, YspE2 and YP_3417 as revealed by our previous comparative proteomic study[32]. To further characterize the secretion of YspE1 and YspE2 by *Y. pestis* strain 201, a biovar Microtus strain, their presence in culture medium of bacteria (Supplementary Table 1) grown under the T3SS-inducing conditions were analyzed by immunoblotting using antibodies raised against recombinant YspE1 and YspE2 proteins (Fig. 1). YscI and YscF are rod and needle proteins, respectively, which have been shown to be essential for T3SS injectisome assembly and substrates translocations[35]. Disruption of T3SS injectisome totally abolished the secretion of YspE1 and YspE2 in both the *ΔyscI* and *ΔyscF* strains, and the *ΔyscI* strain complemented with YscI expressing plasmid restored YspE1 and YspE2 secretions (Fig. 1a, upper panel). Unexpectedly, we found that YspE1 secretion was blocked in *ΔyspE2* although its expression has not been affected (Fig. 1a, lower panel), suggesting that YspE2 might be involved in YspE1 secretion through an as yet uncharacterized mechanism. However, it's not the case for YspE2, which was normally secreted by *ΔyspE1*. In addition, a larger amount of YspE1 and YspE2 secretion occurred in bacteria grown at 37 °C in a chemically defined medium (TMH) without $Ca^{2+}$ (Fig. 1b), consistent with the low calcium response characteristic of *Yersinia* T3SS[36]. Further adenylate cyclase reporter assay confirmed that YspE1 and YspE2 could be delivered into the host cytosol as efficient as YopM, a bona fide T3SS effector (Fig. 1c).

*yspE1* and *yspE2* share more than 90.0% nucleic acid sequence identity and 84.3% amino acid sequence identity and both of them harbor an N-terminal LRR domain and a C-terminal NEL domain (Fig. 2a). BLAST analysis showed that YspE1 and YspE2 are highly similar to a group of NEL domain containing effectors, including *Shigella* IpaH9.8 and IpaH4.5, *Salmonella* SspH2 and SspH1, and SlrP effectors (Fig. 2b). Among them, IpaH9.8 is the closest one with 36.9% and 36.7% amino acid sequence identity of YspE1 and YspE2, respectively (Supplementary Figure 1). It has been reported that *Shigella* IpaH9.8 targets hGBPs for ubiquitination and proteasomal

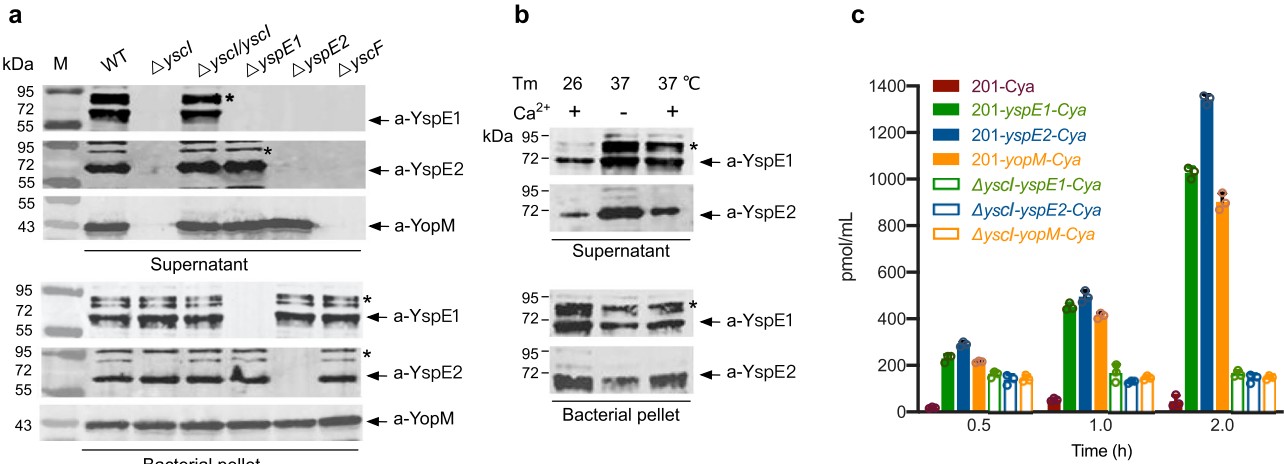

**Fig. 1 | Secretion of NELs of *Y. pestis* Is T3SS Dependent. a** Bacterial strains of the wild type *Y. pestis* and various mutants were grown in TMH medium without calcium at 37 °C and proteins in culture supernatants and bacterial pellets were analyzed by immunoblotting with specific antibodies against YspE1, YspE2 or YopM. The bands labeled with "*" might come from some unknown complexes involved in YspE1/YspE2. **b** *Y. pestis* strain 201 were grown in TMH at 26 °C, or in TMH with or without calcium at 37 °C as indicated. Proteins in culture supernatants and bacterial pellets were analyzed as described in **a**. The image of **a** and

**b** represents results from three independent experiments. **c** *Y. pestis* strain 201 or *ΔyscI* carrying the empty Cya reporter plasmid (201-Cya), or plasmids containing the N-terminal coding sequence of *yspE1* (201- *yspE1-Cya*), *yspE2* (201-*yspE2-Cya*) or *yopM* (201-*yopM-Cya*) fused with *Cya* gene were grown at 26 °C in LB and shifted to 37 °C for 2 h incubation prior to infection. HeLa cells were infected with above strains for different hours and adenylate cyclase activity in the lysate of the infected cells were analyzed. Triplicates samples were analyzed for each reporter strain and the averages with standard deviations were shown.

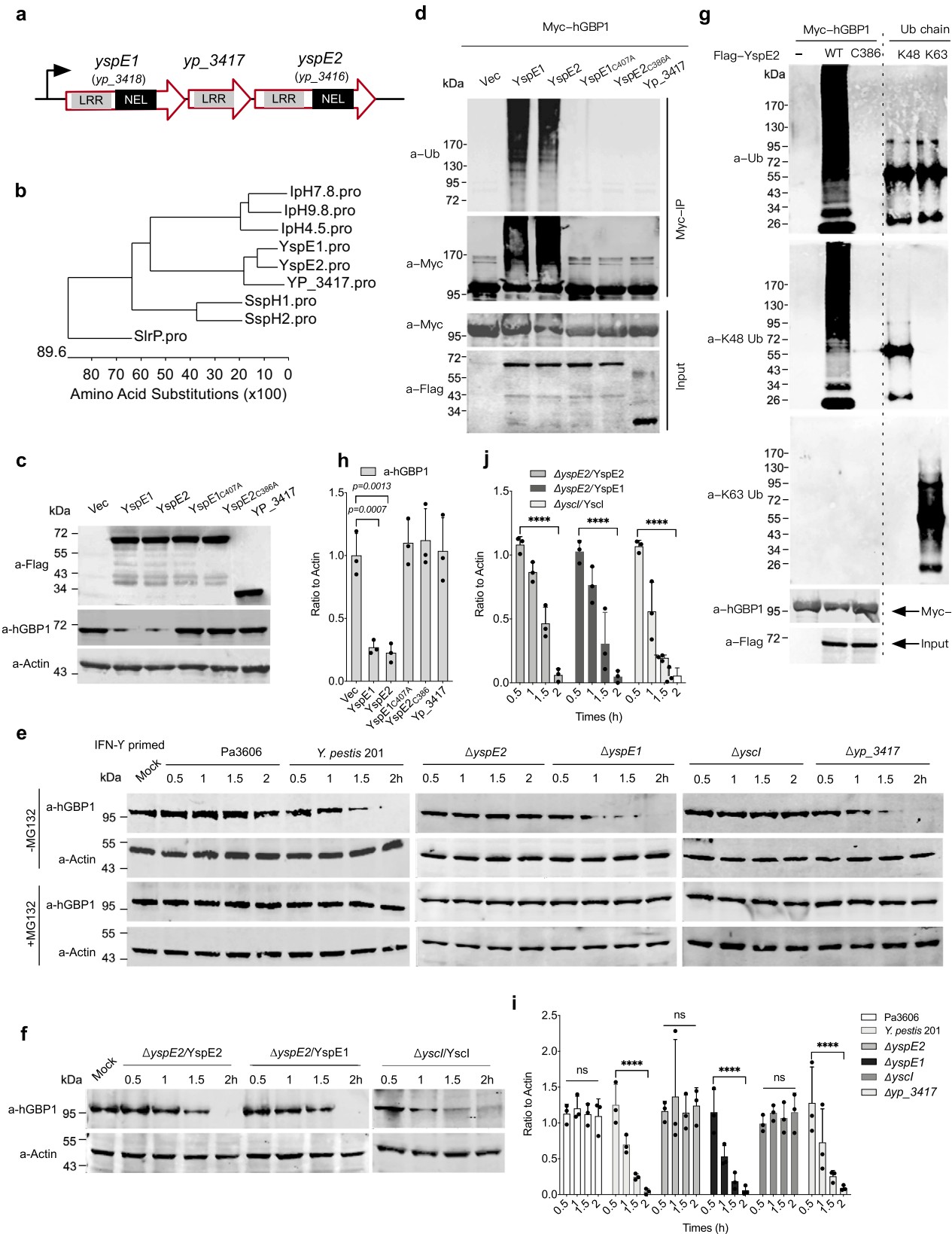

degradation, which ultimately promotes the intracellular replication, motility and spread of *S. flexneri* bacteria[30,31]. To determine whether YspE1 and YspE2 show similar activity to hGBPs, we firstly constructed plasmids expressing YspE1 and YspE2 with an alanine substitution at the critical cystine residue of CEDR motif, YspE1$_{C407A}$ and YspE2$_{C386A}$, which lost the E3 ligase activity. HeLa cells stably expressing GFP-fused

hGBP1 were individually transfected with the plasmids expressing FLAG-tagged YspE1, YspE2 or YP_3417, as well as YspE1$_{C407A}$ and YspE2$_{C386A}$, the corresponding E3 ligase catalytic dead mutants. Expression of either YspE1 or YspE2 induced a significant decrease of hGBP1 in HeLa cells, whereas YspE1$_{C407A}$ and YspE2$_{C386A}$ expressions, as well as YP_3417, showed no such activity (Fig. 2c, h).

**Fig. 2 | YspE2 of *Y. pestis* ubiquitinates hGBP1 and leads to its proteasomal degradation. a** Schematic diagram of the gene loci encoding the NELs of *Y. pestis* (gene IDs are labeled according to the annotation of *Y. pestis* 91001 genome). **b** Sequence analysis of the most closely related NELs from different pathogenic bacteria. **c, d** Empty vector of p3×Flag-CMV and the plasmids expressing YspE1, YspE2 and their ligase catalytic mutants, as well as YP_3417, were transiently co-transfected into 293 T with pCMV-Myc-hGBP1, and hGBP1 was detected by immunoblotting analysis (**c**) or co-immunoprecipitated with anti-c-Myc beads (**d**), followed by immunoblotting analysis using different antibodies. **e** HeLa cells stably expressing GFP-hGBP1 were infected with Pa3606, *Y. pestis* 201 and the various mutants in the presence or absence of MG132 as indicated, and the infected cells were collected at different times and subjected to SDS-PAGE separation and immunoblotting. **f** HeLa cells stably expressing GFP-hGBP1 were infected with *ΔyspE2* complemented with pBAD24-YspE1 or pBAD24-YspE2, *ΔyscI* complemented with *ΔyscI*-pBAD24-YscI, and the infected cells were analyzed as described in **e**. **g** pCMV-Myc-hGBP1 was co-transfected into 293 T cells with p3×Flag-CMV-YspE2 or p3×Flag-CMV-YspE2$_{C386A}$, and hGBP1 was immunoprecipitated and detected using antibodies specific for K63-linked Ub, K48-linked-Ub, hGBP1 and FLAG-tag, respectively. K48-linked and K63-linked ubiquitin chains were run on SDS-PAGE and served as positive controls for immunoblotting detection. The image represents results from two independent experiments. **h–j** represented the quantitative and statistical analysis results of **c, f** and **e** (-MG132 only), respectively, from three independent experiments. Image J was used to obtain the quantitative results and bars represent mean ± S.D. Two-way ANOVA was performed to analyze the significance of difference in protein levels (**$p < 0.01$; ***$p < 0.001$; ****$p < 0.0001$; ns, not significant).

Co-immunoprecipitation assay followed by immunoblotting with the anti-Ub and anti-Myc antibodies showed that immunoprecipitated Myc-tagged hGBP1 were ubiquitinated in cells expressing YspE1/YspE2, but not in those expressing YspE1$_{C407A}$, YspE2$_{C386A}$ or YP_3417 (Fig. 2d).

We further sought to examine the hGBP1 levels during *Y. pestis* infection. *Y. pestis* has evolved clonally from enteropathogenic *Y. pseudotuberculosis* serotype O1:b[3,37]. Thus, we included a serotype O1:b *Y. pseudotuberculosis* strain Pa3606 for hGBP1 degradation analysis. IFN-γ treated HeLa cells stably expressing hGBP1 were infected with the wild type *Y. pestis* strain 201 or Pa3606, alongside the various mutants of *Y. pestis* including *ΔyspE1*, *Δyp_3417*, *Δ yspE2* and *ΔyscI*. The decrease of hGBP1 became apparent at one hour post-infection (hpi) in 201-infected HeLa cells, and this effect was abrogated in *ΔyscI* or *ΔyspE2*, but not in *ΔyspE1* infected HeLa cells (Fig. 2e, i). It is reasonable since the secretion of both YspE1 and YspE2 was blocked in *ΔyspE2*; however, *ΔyspE1* secrets YspE2 normally (Fig. 1a). Furthermore, addition of MG132 that is a potent cell-permeable 20 S proteasome inhibitor totally abolished hGBP1 reduction, indicating that hGBP1 was degraded via proteasome-mediated degradation pathway. Consistent with the above results showing that the secretion of YspE1 and YspE2 are T3SS dependent, blockage of T3SS abrogated the ability of *Y. pestis* to degrade hGBP1 (Fig. 2e), and complementing *ΔyscI* with YcsI restored the hGBP1 degradation phenotype (Fig. 2f, j). These results indicated that *Y. pestis* infection led to hGBP1 ubiquitination and degradation in a YspE1/YspE2 and E3 ligase activity dependent manner, implying that they could target hGBPs as well in a similar way to *Shigella* IpaH9.8[30,31]. Plasmids expressing FLAG-tagged YspE2 or YspE2$_{C386}$ and Myc-hGBP1 were co-transfected into 293 T cells and the immunoprecipitated hGBP1 were analyzed and found to be ubiquitinated via K48 but not K63 (Fig. 2g), further confirming that hGBP1 was degraded via K48-linked ubiquitination and proteasomal degradation pathway. Interestingly, hGBP1 was quite stable in Pa3606-infected HeLa cells and no degradation has been detected during infection. Sequence alignment analysis found that both the counterparts of YspE1/YspE2 E3 ligase in Pa3606 genome harbor only LRR domains but no NEL domain, explaining well why Pa3606 fails to degrade hGBP1.

Given that the secretion of both YspE1 and YspE2 was blocked when *yspE2* was deleted, *ΔyspE2* complemented individually with the plasmids expressing YspE1 or YspE2 were constructed and used to infect the HeLa cells. The infected cells were then subjected to hGBP1 degradation analysis, and the results found that expressing either of them restored the hGBP1 degradation capability (Fig. 2f). Taken together, our results showed that both YspE1 and YspE2 are able to ubiquitinate hGBP1 through K48-linked ubiquitination, leading to the hGBP1 degradation via the ubiquitin-proteasomal pathway.

### *Y. pestis* infection leads to degradation of multiple hGBPs while *Y. pseudotuberculosis* is unable to dampen this defense mechanism

There are 7 hGBPs encoded in the human genome. To determine whether *Y. pestis* 201 infection impacts hGBPs other than hGBP1, HeLa cells stably expressing GFP-fused hGBP1-7 were individually infected with *Y. pestis* 201, and the degradation of hGBPs was analyzed by direct fluorescence microscopy and immunoblotting. The abundance of hGBP2, 5 and 6, but not 3, 4 and 7, significantly decreased in 201-infected cells, indicating that multiple hGBPs can be targeted for degradation by *Y. pestis* (Fig. 3a and d) and the results from fluorescence microscopic observation further confirmed these findings (Fig. 3b). Deletion of *yspE2* abrogated the ability of *Y. pestis* for hGBPs degradation (Supplementary Fig. 2a), which were well restored by complementation with YspE1 or YspE2 (Supplementary Figure 2b). MG132 treatment totally abolished the reduction of hGBP2, 5 and 6, indicating they were also degraded via the ubiquitin-proteasomal pathway in a similar way to hGBP1 (Supplementary Fig. 2a). Furthermore, the endogenous hGBP1 was degraded by 201 but not by *ΔyspE2* in IFN-γ-primed HeLa and U937 cells at 2 hpi (Fig. 3c, e), suggesting a potential capability of *Y. pestis* to degrade GBPs during infection of host.

In searching of the E3 ligase homologs in *Y. pseudotuberculosis*, we found that although there are homologs of *yspE1* and *yspE2* in some strains, they are often incomplete and harbor only one of the genes. Besides, *yspE1* homologs in different *Y. pseudotuberculosis* strains are highly diversified in sequences (total of 23 *Y. pseudotuberculosis* strains with complete genomes available in the NCBI database were analyzed) (Fig. 4a, Supplementary Table 3). By contrast, 28 *Y. pestis* genomes (total of 38 *Y. pestis* strains with complete genomes available in the NCBI database were analyzed) encode exactly identical YspE1, while the 9 remaining strains (except for 1 strain of branch 0.ANT3) belonging to the branch 0.PE (Pestoides), which are deemed to be intermediate between modern *Y. pestis* isolates and *Y. pseudotuberculosis*, contain no *yspE1/yspE2* homologs at all (Fig.4b, Supplementary Table 2). The similar patterns could also be found in the heatmaps of gene *yp_3417* and *yspE2* (Supplementary Fig. 3), except for two isolates of *Y. pestis* 195 P and Nepal516 (both of them contains an functional *yspE1*), having a relative low identity in the *yspE2* gene with other *Y. pestis* stains (Supplementary Fig. 3b).

According to the homology alignment results, we selected 3 representative *yspE1* homologs in *Y. pseudotuberculosis* strains, i.e. IP2666pIB1, YPIII and PB1 + (Supplementary Fig. 4a), and cloned them into p3×Flag-CMV vector, as well as their corresponding *yspE2* homologs in the first two strains because PB1 + does not contain a *yspE2* homolog. The constructed plasmids expressing YspE1/2 and its' homologs were transfected into HeLa cells stably expressing GFP-fused hGBP1-7. Immunoblotting analysis using anti-hGBPs antibodies found that expression of YspE1 and YspE2 led to the significant degradation of hGBP 1, 2, 5 and 6; however, none of the YspE1 (Fig. 4c, h and Supplementary Fig. 4b) and YspE2 homologs (Fig. 4 d, i and Supplementary Fig. 4c) in IP2666pIB1, YPIII and PB1 + could degrade any of hGBPs.

To further explore the reasons leading to the failure of these YspE1/2 homologs in GBPs degradation, we constructed a serials of plasmids expressing chimeric proteins by swapping the LRR domain and NEL

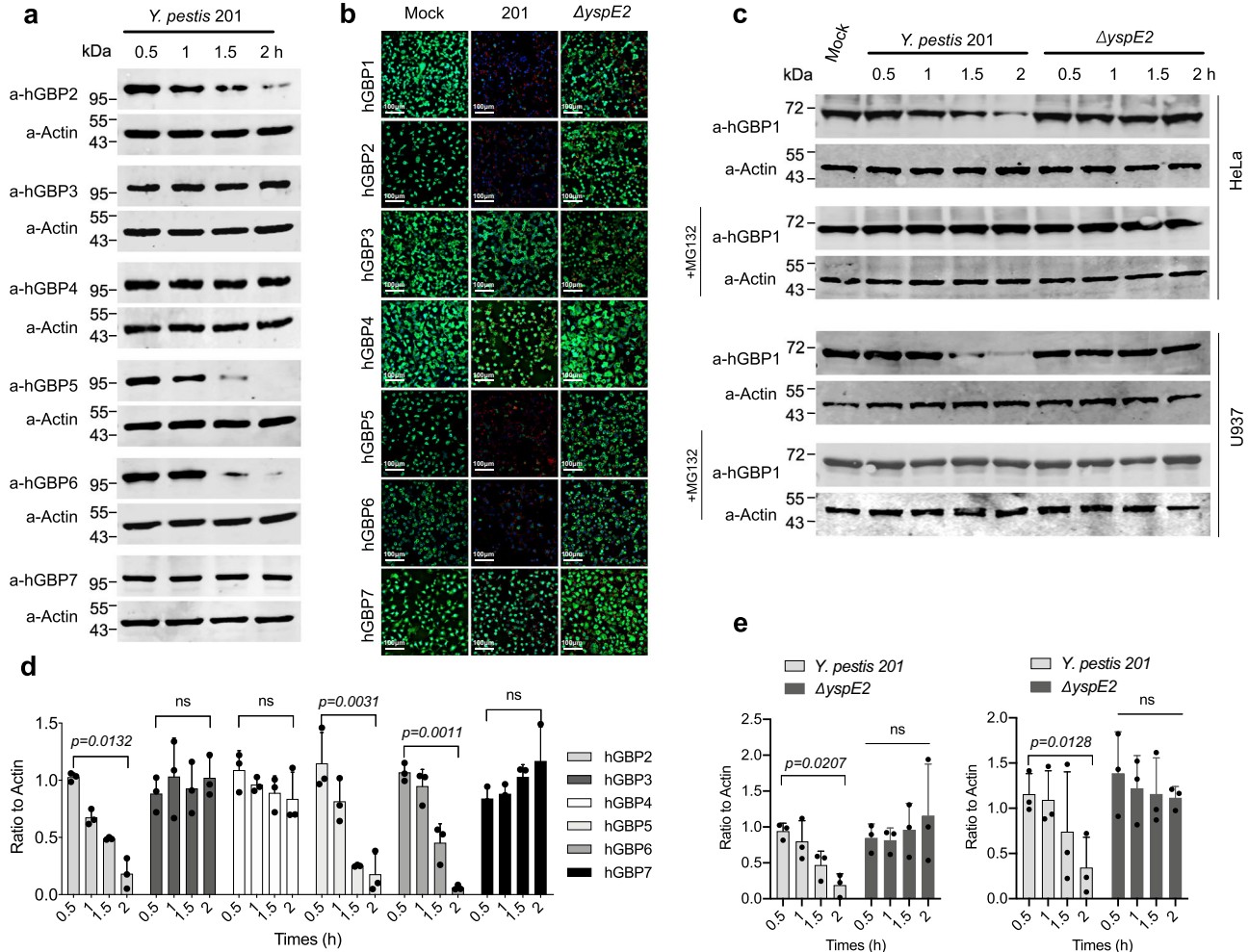

**Fig. 3 | *Y. pestis* degrades multiple hGBPs during infection. a** HeLa cells stably expressing GFP-hGBP1-7 were infected with *Y. pestis* 201 and hGBPs in samples collected at different times were immunoblotted with specific antibodies. **b** HeLa cells stably expressing GFP-hGBP1-7 were infected with *Y. pestis* or *ΔyspE2*, and confocal florescence images were taken at 2.5 hpi. The images represent results from three independent experiments. **c** Endogenous GBP1 degradation in HeLa and U937 cells. **d, e** Represented the quantitative and statistical analysis results of **a** and **c**, respectively, from three independent experiments. Image J was used to obtain the quantitative results and bars represent mean ± S.D. Two-way ANOVA was performed to analyze the significance of difference in protein levels (*$p < 0.05$; **$p < 0.01$; ns, not significant).

domain between the homologous protein pairs. HeLa cells stably expressing GFP-fused hGBP1 was transfected with plasmids expressing various chimeric proteins, and the level of hGBP1 was then determined. The fused protein of YspE1-LRR domain and the NEL domain of YspE1$_{IP2666}$ (Fig. 4e, j), YspE1$_{YPIII}$ (Fig. 4f, k) and YspE1$_{PB1+}$ (Fig. 4g, l) can degrade hGBP1 as efficiently as YspE1, whereas the fused protein of YspE1-NEL domain and the LRR domain of YspE1$_{YPIII}$, YspE1$_{IP2666}$ and YspE1$_{PB1+}$ showed no impacts on hGBP1 level. Similar results were obtained for YspE2 domain-swapping experiments (Fig. 4e, f, j, and k). These results demonstrated that the sequence variations in the LRR domain, instead of the NEL domain, resulted in the failure of hGBP1 degradation by YspE1$_{PB1+}$, YspE1/2$_{IP2666}$ and YspE1/2$_{YPIII}$.

**YspE1 and YspE2 are functionally redundant in degradation of both hGBPs and mGBPs**

Above results shown that *Y. pestis* infection mediated hGBP1, 2, 5 and 6 degradations via ubiquitin-proteasome pathway. To identify the E3 ligase activity of YspE1 and YspE2 towards the individual hGBPs, p3×FLAG-CMV-YspE1 and -YspE2 were transiently transfected into 293 T cells with the Myc-hGBPs expressing plasmids, and the degradation of specific hGBPs were determined by immunoblotting (Fig. 5a–f, r). In agreement with the cell infection assay results shown in

Fig. 3a, ectopic expression of either YspE1 or YspE2 led to the specific degradation of hGBP1, 2, 5 and 6, suggesting that the two E3 ligases have the identical substrates preference and are functionally redundant for hGBPs degradations. The specificity of NEL family E3 ligase relays on the binding of LRR domain to their substrates, which is the prerequisite for exerting ubiquitination activity as well[33]. Thus, co-immunoprecipitation assays were performed to analyze the binding capability between the protein pairs. Both the full length YspE1 or YspE2 and their LRR domains exhibited the binding affinity to hGBP1, 2, 5 and 6 (Supplementary Fig. 5a–d) but not to hGBP3, 4 and 7 (Supplementary Fig. 5e–g). YP_3417 that contains only LRR domain can also bind to hGBP1, 2, 5 and 6 (Supplementary Fig. 5a–d), suggesting that it might play roles in regulation of the targeting ubiquitination of host proteins by competitive binding to the substrates of YspE1/E2.

To exclude the possibility that hGBPs are degraded via an indirect mechanism dependent on YspE1/E2, FLAG-tagged YspE2 or YspE2$_{C386A}$ expressing plasmids were individually co-transfected into 293 T cells with Myc-tagged hGBP1-7 expressing plasmids, followed by co-immunoprecipitation and immunoblotting with the anti-Ub and anti-Myc antibodies. The poly-ubiquitinated proteins were successfully detected for hGBP1, 2, 5 and 6, but not for hGBP3, 4 and 7 in the transfected 293 T cells (Fig. 5g). Mutation of Cys386 to alanine in the

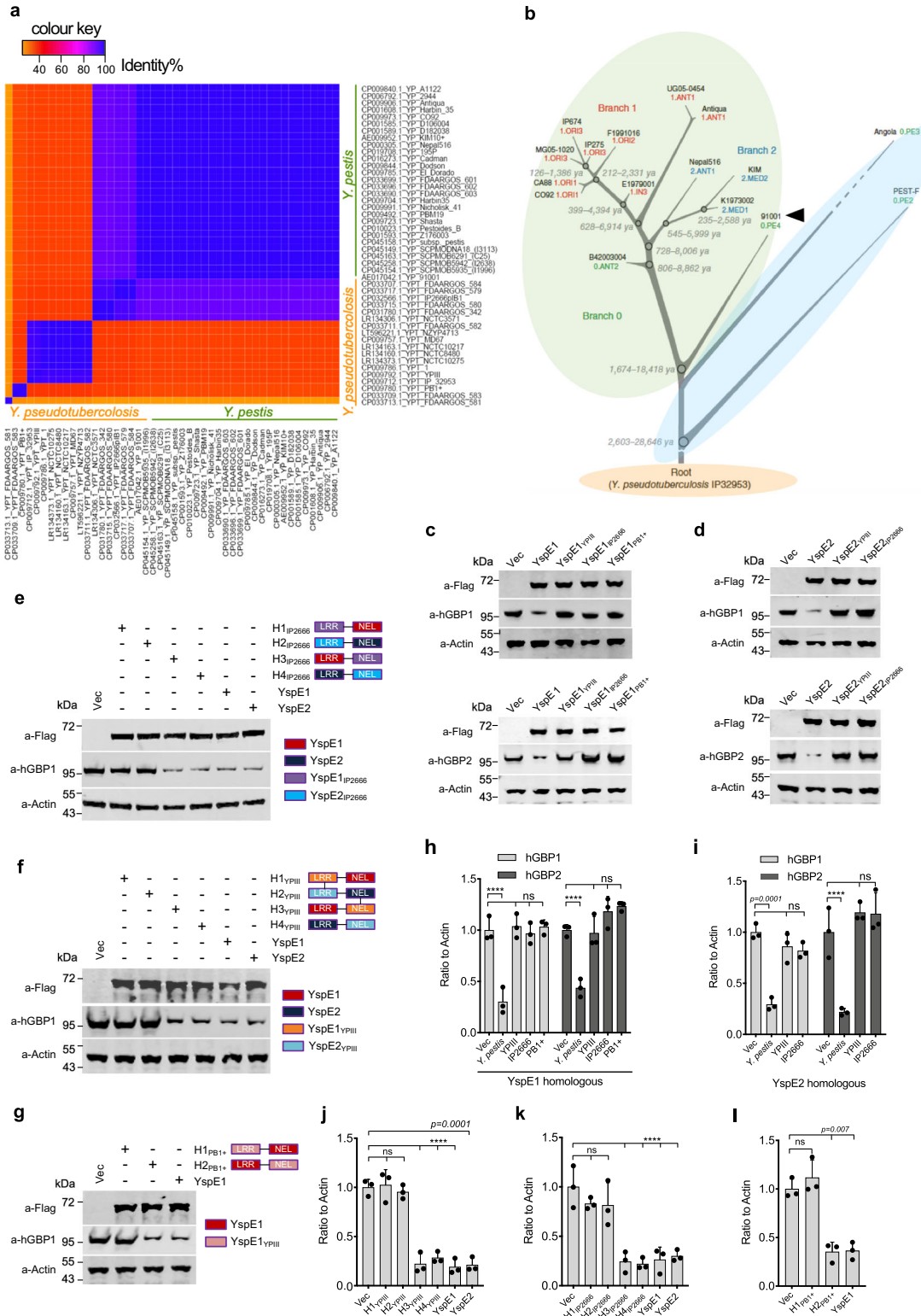

NEL domain of YspE2 diminished the ubiquitinated products of hGBPs (Fig. 5h), indicating that the E3 ligase activity is responsible for ubiquitination of hGBP1, 2, 5 and 6. These results demonstrated that YspE1/E2 can directly ubiquitinate hGBPs in the host cells.

Full-length YP_3417, YspE1, YspE2 proteins and their LRR domains immobilized on standard surface plasmon resonance (SPR) showed a dose dependent resonance signal and rapid association with hGBP1 (Fig. 5, i–l), while did not bind to hGBP4 that served as a negative control (Fig. 5m, n). The calculated dissociation constants ($K_D$)

between YspE1/YspE2 and hGBP1 were $3.7 \times 10^{-7}$M and $1.5 \times 10^{-6}$M, respectively, and the LRR domain of YspE1/YspE2 showed the similar $K_D$ to their full-length counterparts, all of which are lower than that of *Shigella* IpaH9.8 ($2.9 \times 10^{-8}$M). $K_D$ between YP_3417 and hGBP1 was the lowest one in all the tested molecules mainly due to a relatively higher dissociation rate constant (Fig. 5o).

There are eleven GBPs known in mouse, encoded on the chromosome 3 (mGBP 1, 2, 3, 5 and 7) and 5 (mGBP4, 6, 8, 9, 10 and 11), respectively. We further determined how the stability of mGBPs can be

**Fig. 4 | YspE1/YspE2 homologs in *Y. pseudotuberculosis* cannot degrades hGBPs. a** Heatmap of intra- and inter species sequence identity for *yspE1* homologs among *Y. pestis* or *Y. pseudotuberculosis* strains with complete genomes available in the NCBI database. The identity values are shown in a color scale ranging from orange to blue, with purple and red in between. **b** Phylogenic tree (adapted from reference [2]) labeled with different colors according to the presence of gene homologs of *yspE1/yspE2*. *Y. pestis* strains within green ellipse encode homologous proteins nearly identical to YspE1 and strains within blue ellipse completely lost the homolog of *yspE1/yspE2* gene locus. The biovar Microtus *Y. pestis* 91001 was labeled with an arrow head. GBPs degradation activity of YspE1 (**c**) and YspE2 (**d**) homologs in *Y. pseudotuberculosis* IP2666pIB1, YPIII and PB1 + . Plasmids expressing FLAG-tagged YspE1/YspE2 homologs in three representative *Y. pseudotuberculosis* were individually transfected into IFN-γ primed HeLa cells stably expressing GFP-tagged hGBP1 or hGBP2 as indicated and the transfected cells were lysed and immuno-blotted with anti-hGBP antibodies. Domain-swapping experiments between YspE1/YspE2 and their homologs in IP2666pIB1 (**e**) and YPIII (**f**). **g** Domain-swapping experiments between YspE1 and it homolog in PB1 + . Plasmids expressing hybrid proteins by swapping the LRR domain and NEL domain between the homologous protein pairs as indicated were transfected into HeLa cells expressing GFP-tagged hGBP1 and the level of hGBP1 was measured by immunoblotting. **h, i** represented the quantitative and statistical analysis results of **c** and **d**, respectively, from three independent experiments. **j–l** Represented the quantitative and statistical analysis results of **e**, **f** and **g**, respectively, from three independent experiments. Image J was used to obtain the quantitative results and bars represent mean ± S.D. Two-way ANOVA was performed to analyze the significance of difference in protein levels (***$p < 0.001$; ****$p < 0.0001$; ns, not significant).

affected by YspE1 and YspE2. Co-transfection assay results in 293 T cells showed that expression of YspE1 or YspE2 resulted in a significant decrease of mGBP1, 3, 6, 7, 10 and 11 and no different sub-strate preference was found between YspE1 and YspE2 (Fig. 5p, s). As expected, expression of YP_3417 showed no such activity to any of the mGBPs. The endogenous mGBP1 was degraded by 201 but not by *ΔyspE2* in IFN-γ-primed RAW264.7 cells at 2 hpi (Fig. 5q, t). Taken together, our results demonstrated that YspE1 and YspE2 are functionally redundant and the substrates specificity of YspE1 and YspE2 were totally identical with each other for both hGBPs and mGBPs.

## YspE1/2 disrupt co-localization of hGBP1 with the intracellular *Y. pestis* and promote the bacteria survival during infection of HeLa cells

hGBPs are critical for host defense against the intracellular bacteria[26,38] and hGBP1 has been shown to be colocalized with the bacteria inside the cells[30,31]. To analyze the impact of hGBP1 ubiquitination on infec-tions caused by *Y. pestis*, the pUC19-Scarlet plasmid was transformed into the bacterial strains to make them express red fluorescent protein (RFP). IFN-γ primed HeLa cells stably expressing hGBP1 were infected with the different RFP-expressing *Y. pestis* strains grown at 26 °C for facilitating the internalization of bacteria, and confocal micrographs of the infected cells were taken at 1 hpi (Fig. 6a). There was a strong positive correlation between RFP and GFP signal intensities in *ΔyspE2*-infected cells, indicating a significant co-localization of hGBP1 with *Y. pestis ΔyspE2*. By contrast, much less co-localization signals was found for the wild-type bacteria, probably due to the degradation of hGBP1 by 201. Expressing YspE2, but not YspE2$_{C386A}$ in *ΔyspE2* diminished the colocalization of bacteria with hGBP1 (Fig. 6a). Pearson's correlation coefficient values for co-localization of hGBP1 and wild type *Y. pestis*, *ΔyspE2*, *ΔyspE2* complemented with YspE2 or YspE2$_{C386A}$ were 0.19, 0.77, 0.20 and 0.80, respectively (data not shown). GBPs colocalized with cytosolic bacteria has been shown to participate in disruption of bacterial integrity and promote the clearance of invading microorganisms[28,30,31]. To investigate how the hGBP1 colocalized with *ΔyspE2* observed in the IFN-γ primed HeLa cells influences the fate of the bacteria, we performed live microscopy analysis. Upon infection, GFP-hGBP1 signals in HeLa cells infected with the wild type *Y. pestis* attenuated gradually (Fig. 6b, Supplementary Movie 1). However, in the *ΔyspE2*-infected cells, GFP intensity was quite stable and RFP-expressing *ΔyspE2* bacteria decreased dramatically (Fig. 6b, Supple-mentary Movie 2), indicating that a significant amount of bacteria were killed after taken up into the HeLa cells. Complementing of *ΔyspE2* with YspE2, but not YspE2$_{C386A}$, restored the ability of bacteria to degrade hGBP1 (Supplementary Movies 3 and 4). These results demonstrated that hGBPs mediated cell-autonomous antimicrobial defenses contribute greatly to the restriction of *Y. pestis ΔyspE2* infection by HeLa cells, whereas the wild type *Y. pestis* can evade this mechanism by utilizing YspE1/2 to degrade hGBPs.

## YspE1/2 promote the survival of *Y. pestis* in macrophages and inhibit inflammasome activation

Murine GBPs are functional homologs to human hGBPs. We gen-erated *Gbp^chr5−/−* mice by removing the locus containing mGBP4, 6, 8, 9, 10 and 11, and *Gbp^chr5−/−* were crossed with *Gbp^chr3−/−* mice[31] to generate *Gbp^chr3−/−,chr5−/−* mice. The bone marrow derived macro-phages (BMDMs) isolated from the wild type (Fig. 7a, b) and the *Gbp^chr3−/−,chr5−/−* mice (Fig. 7d, e) were left untreated or were primed with IFN-γ, followed by infection with the *Y. pestis* 201 or *ΔyspE2*, and bacterial survival percentage was analyzed by plating and counting of living bacteria released from BMDMs. A significant replication has been found for *Y. pestis* 201 ($p < 0.0001$) at 2 hpi in comparison to *ΔyspE2* (Fig. 7a, c). Besides, *Y. pestis ΔyspE2* was cleared much more rapidly ($p < 0.0001$, at 4 hpi) than 201 in the wild type BMDMs (Fig. 7a, c), indicating that YspE1/2 significantly promote the replication and survival of *Y. pestis* in macrophages. IFN-γ pre-stimulation greatly promoted the killing of *Y. pestis* bacteria by the wild type BMDMs for both *Y. pestis ΔyspE2* and 201 ($p < 0.0001$) (Fig. 7b, c), consistent with the established function of IFN-γ against plague infection[19,20]. In contrast, no significant dif-ference in survival in *Gbp^chr3−/−,chr5−/−* BMDMs was found between 201 and *ΔyspE2* (Fig. 7d, f). The enhancement effect in bacterial killing of IFN-γ was also not significant in *Gbp^chr3−/−,chr5−/−*-BMDMs at 2 hpi for both strains; however, the number of the living 201 bacteria was significantly higher than *ΔyspE2* at 4 hpi (Fig. 7e, f), indicating that IFN-γ induced cellular targets of YspE1/2 other than GBPs might be present and plays roles in *Y. pestis* pathogenesis. Remarkably, replication of both strains in *Gbp^chr3−/−,chr5−/−* BMDMs were higher than those in wild type BMDMs, underlying the importance of GBPs in restriction of intracellular bacteria (Fig. 7g). These data con-firmed that GBPs are critical for intracellular bactericidal activity of macrophages to *Y. pestis* and GBPs degradations mediated by YspE1/YspE2 significantly promote survival of *Y. pestis* in macrophages.

GBPs has been shown to play crucial roles in multiple process of inflammasome activation[39–41]. IFN-γ-primed human macrophages-like U937 cells were infected with various *Y. pestis* strains and the infected cells were lysed at 4 hpi and both culture supernatants and cell lysates were subjected to detection of IL-1β, IL-18 and p20, the active form of caspase-1. Maturated IL-1β and caspase-1 p20 secretions by *ΔyspE2*-infected cells were much higher than those by 201-infected cells, suggesting that inflammasome activation and IL-1β maturation were strongly inhibited by *Y. pestis* and YspE1/YspE2 was required for this process (Fig. 7h, k). Similar inhibitory effects on inflammasome acti-vation could be observed in HeLa cells infected with the wild type *Y. pestis* (Supplementary Fig. 6).

Similarly, in RAW264.7 and the wild type BMDMs infection assays, only *ΔyspE2* infection led to the inflammasome activation and the rest *Y. pestis* strains producing YspE1/2 E3 ligase totally abrogated the activation process (Fig. 7i, l). Notably, due to low levels of caspase-1

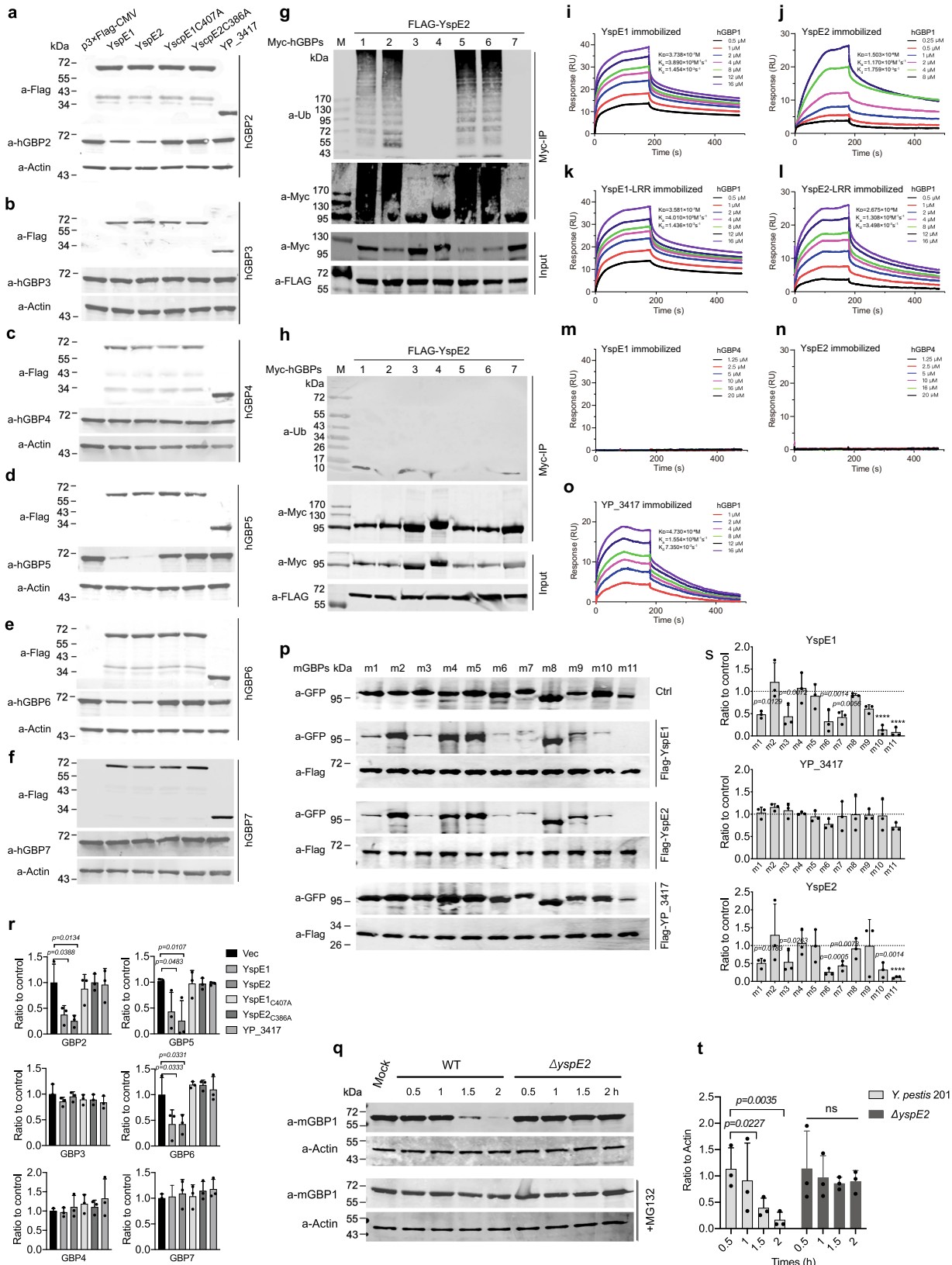

p20 in BMDMs-*Gbp*[chr3–/–,chr5–/–] cells, we were unable to detect significant inflammasome activation in these cells during infection with the *ΔyspE2*, as well as the strains producing YspE1/YspE2 (Fig. 7j, l), suggesting GBPs are required for *Y. pestis* induced inflammasome activation.

## YspE1/2 degradation of GBPs play a significant role in disease progress in a mouse model of bubonic plague

*Y. pestis* is typically transmitted by fleas and causes bubonic plague. To assess the function of YspE1/2 degradation of GBPs in the development of bubonic plague, mice were subcutaneously (*s.c*) challenged with

**Fig. 5 | Analysis of binding and ubiquitination activities of YspE1/YspE2 to hGBPs. a–f** In vivo binding assay between YspE1/YspE2/YP_3417 to the different hGBPs. Plasmids expressing FLAG-tagged YspE1, YspE2, YspE1$_{C407A}$, YspE2$_{C386A}$ and YP_3417 were individually transfected into 293 T cells as indicated and the trans-fected cells were lysed and immunoblotted with anti-hGBP antibodies. Ubiquiti-nation assay of hGBPs by YspE2. pCMV-Myc-hGBP1-7 plasmids were individually co-transfected with p3×FLAG-YspE2 (**g**) or -YspE2$_{C386A}$ (**h**), and the total cell extracts were immunoprecipitated with anti-c-Myc beads and immunoblotted with anti-Ub antibody. The image of **g** and **h** represents results from at least three independent experiments. **i–l**, **o** Binding affinity between various *Y. pestis* proteins and hGBP1 were analyzed by SPR and hGBP4 served as a negative control (**m**, **n**). **p** Plasmids expressing GFP-tagged mGBP1-11 were individually co-transfected with FLAG-tagged YspE1, YspE2 or YP_3417 into 293 T cells. Different mGBPs were co-immunoprecipitated with FLAG-beads, subjected to SDS-PAGE and immunoblot-ting analysis using anti-GFP and anti-FLAG antibodies. **q** Endogenous GBP1 in RAW264.7 infected with the wild type *Y. pestis* 201 or *ΔyspE2*. **r–t** represented the quantitative and statistical analysis results of **a–f**, **p** and **q** respectively, from three independent experiments. Image J was used to obtain the quantitative results and bars represent mean ± S.D. Two-way ANOVA was performed to analyze the sig-nificance of difference in protein levels (*$p < 0.05$; **$p < 0.01$; ***$p < 0.001$; ****$p < 0.0001$; ns, not significant). Only the statistically significant differences were labeled in (R).

-100 CFU of different strains (Fig. 8a, b). Mice infected with *ΔyspE2* survived three more days on average than those of 201-infected group, and expressing either YspE1 or YspE2 in *ΔyspE2* restored the virulence to mice (Fig. 8a), indicating that lacking of E3 ligase led to a significant virulence attenuation (Log-rank test, $p < 0.001$). The numbers of *ΔyspE2* from groin lymph node, spleen, lung and liver were one to two orders of magnitude lower than that of 201 strain, and complementing *ΔyspE2* with YspE1 or YspE2 restored the colonization ability of bacteria in these tissues (Fig. 8b). These results showed that YspE1/2 degradation of GBPs play a significant role in the development of bubonic plague, and further confirmed that YspE1 and YspE2 are not only functionally redundant in degradation of GBPs, but also in systemic plague infection.

### *Gbp*$^{chr3−/−,chr5−/−}$ mice are more susceptible to *Y. pestis* infection

To test the roles of GBPs in the fight against *Y. pestis* infection, *Gbp*$^{chr3−/−,chr5−/−}$ and C57BL/6 J mice were *s.c* challenged with *Y. pestis* 201 or *ΔyspE2*, respectively. *Gbp*$^{chr3−/−,chr5−/−}$ mice were much more susceptible to *ΔyspE2* than C57 mice with a 4-day reduction in median survival time, indicating that the virulence of *ΔyspE2* has restored in *Gbp*$^{chr3−/−,chr5−/−}$. Meanwhile, the average survival days of *Gbp*$^{chr3−/−,chr5−/−}$ mice was one day less than that of C57 mice after challenged with the similar dose of *Y. pestis* strain 201. Significance of differences between survival curves of 201 and *ΔyspE2* in C57 ($p < 0.0001$) were greatly decreased but still present in *Gbp*$^{chr3−/−,chr5−/−}$ mice ($p = 0.036$), implying again targets other than GBPs of YspE1/E2 exist in host (Fig. 8c). Consistently, the number of 201 bacteria was significantly higher in groin lymph nodes in *Gbp*$^{chr3−/−,chr5−/−}$ ($p < 0.05$) compared to C57 mice and also greatly increased in spleen, lung and liver, although not significant. By contrast, the number of *ΔyspE2* was sig-nificantly higher in all types of tissue in *Gbp*$^{chr3−/−,chr5−/−}$ mice. No sig-nificant differences was found between 201 and *ΔyspE2* bacterial loads in tissues of *Gbp*$^{chr3−/−,chr5−/−}$ mice (Fig. 8d). Taken together, these results demonstrated that GBP-mediated host defense is important for the fight against *Y. pestis*, and the virulence attenuation of *ΔyspE2* in mice was GBP-dependent.

### Discussion

hGBPs are IFN-γ induced proteins that are important in cell-autonomous defense and inflammasome activations[41,42]. It has been well established that *Y. pestis* bacteria are readily engulfed by profes-sional phagocytes when initially invading the host, but rapidly devel-oped phagocytic-resistance by expressing a set of virulence factors and initiate an extracellular lifecycle[36,43]. Therefore, survive and repli-cate within host macrophages is a critical step for *Y. pestis* to establish a systematic infection via flea-biting[44]. In this study, we demonstrated that YspE1 and YspE2 of *Y. pestis* can be translocated by T3SS into host cytosol where they ubiquitin and degrade multiple GBPs via protea-some pathway. GBPs including hGBP1, 2, 5 and 6, as well as mGBP1, 3, 6, 7 10, 11 are degraded by *Y. pestis* in an YspE1/YspE2 dependent manner (Fig. 2c, Fig. 3a, b and Fig. 5p). GBPs degradation by YspE1/2 sig-nificantly promote the survival of *Y. pestis* in both the IFN-γ pre-

stimulated and unstimulated BMDMs (Fig. 7a–c). We observed about only 2-fold replication of the wild type 201 in BMDM at 2 hpi, followed by the rapid clearance of *Y. pestis* in this study, which is inconsistent with the previous finding that the number of living *Y. pestis* increased by 4-14 folds in macrophages over time post-infection[21,22,45]. We spec-ulate that the difference existing in survival curves might originate from the different strains used in these studies (EV766, a biovar Orientalis strain and KIM10$^+$, a biovar Mediaevalis strain versus 201, a biovar Microtus strain), or some different details in experimental procedures. Nevertheless, our data clearly demonstrate a significant role of GBPs degradation by YspE1/2 for *Y. pestis* survival in macro-phages. *Gbp*$^{chr3−/−,chr5−/−}$ BMDMs are more susceptible for *Y. pestis* infection and a significantly enhanced replication of both the wild type and a *yspE2* mutant of *Y. pestis* appeared at 2 hpi in these cells no matter primed with IFN-γ or not. *Y. pestis ΔyspE2* but not the wild type was found to be colocalized with hGBP1 due to the degradation by YspE1/2, and live microscopy analysis indicated GFP-hGBP1 remained stable during infection and the number of *ΔyspE2* decreased sig-nificantly, in sharp contrast to the experimental phenomena observed in the cells infected with the wild type *Y. pestis* that GFP-hGBP1 atte-nuated rapidly post infection, accompanied with high numbers of bacteria (Fig. 6).

*Y. pestis* infection strongly inhibits the host immune responses and Yops effectors that are translocated into the host cytosols have been implicated in these processes[46–48]. Recognition of microbial pathogens by nucleotide-binding oligomerization domain leucine-rich repeat proteins (NLRs) triggers the assembly of inflammasomes and caspase-1 activation. *Y. pestis* inhibits activation of NLRP3, NLRC4 and Pyrin inflammasome using T3SS effectors including YopM, YopJ and YopK via various mechanisms, suggesting that inflammasome is intensively targeted by the arsenal of this pathogen[49–52]. Our data showed that GBP degradation by YspE1/2 significantly inhibit the activation of inflammasomes of both human and murine macro-phages, well in line with the established roles of GBPs in inflammasome activation[39,40,42]. Inhibition of inflammasome activation from several lines contribute synergically to dampening of the host inflammatory response during the initial replication of *Y. pestis* inside the macrophages.

*Y. pestis* has recently evolved from its ancestor *Y. pseudotubercu-losis*, an enteropathogen causing self-limited gastrointestinal disease[3,10], and *Y. pseudotuberculosis* serotype O:1b is thought to be the direct evolutionary ancestor of *Y. pestis*[3,37]. Comparative genomics studies between the two pathogens showed that *Y. pestis* has under-went a genome reductive evolution because some dozens of chro-mosomal genes and two *Y. pestis*-specific plasmids (pPCP1 and pMT1) represent the only new genetic material of *Y. pestis* since the diver-gence from *Y. pseudotuberculosis*, but as many as 13% of *Y. pseudotu-berculosis* genes became pseudogenes or absent at all in *Y. pestis*[4]. Phylogenetic analysis of the *yspE1-yspE2* gene locus, including *yspE1*, *yp_3417* and *yspE2*, in genomes of *Y. pestis* and *Y. pseudotuberculosis* revealed that nearly all the typical *Y. pestis* strains, except for the branch 0.PE (Pestoides) strains deemed to be intermediate between

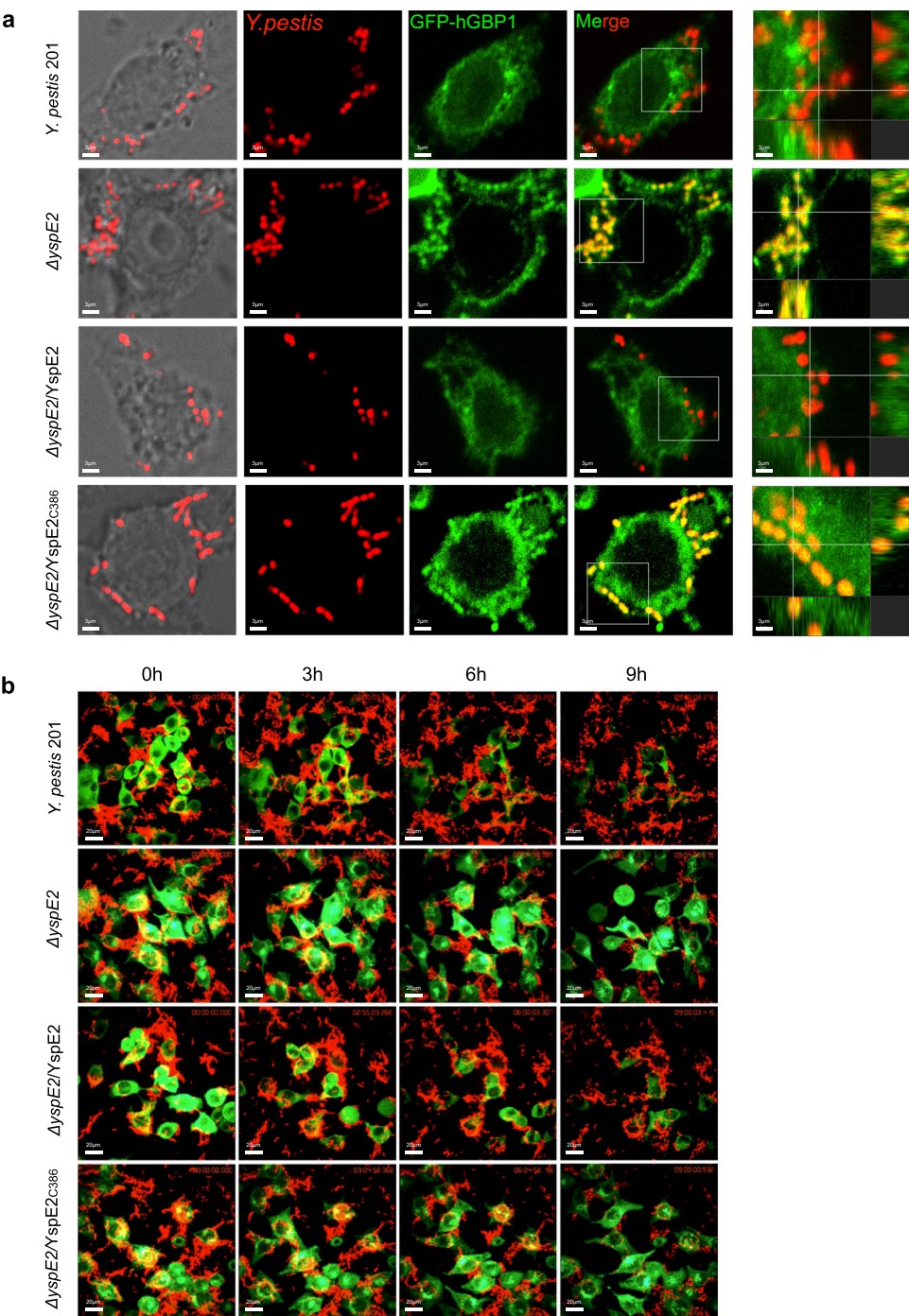

**Fig. 6 | Co-localization of *Y. pestis* Δ*yspE2* with hGBP1 dramatically promotes the clearance of bacteria during infection of HeLa cells. a** HeLa cells stably expressing GFP-hGBP1 were infected with the RFP-expressing wild type *Y. pestis* 201, Δ*yspE2*, Δ*yspE2* complemented with the plasmids expressing YspE2 or YspE2$_{C386A}$ at an MOI of 10. After 1 h of infection, the confocal micrographs were acquired using an Andor Dragonfly spinning disk confocal microscope. The far right column showed the 3D reconstruction images from the multiple slice images of the enlarged framed areas. The intracellular bacteria of *Y. pestis* Δ*yspE2*, but not 201,

was significantly colocalized with hGBP1. **b** Representative frames from Supplementary movie 1 to 4 of HeLa cells expressing GFP-hGBP1 infected with wild type, Δ*yspE2*, Δ*yspE2*/YspE2 or Δ*yspE2*/YspE2$_{C386A}$ RFP-expressing *Y. pestis*. HeLa cells were seeded in Lab-Tek chambered cover-glass and infected with the indicated *Y. pestis* strains and the images were taken every 5 mins until 9 h of infection using spinning disk system with Nikon ECLIPSE Ti-E microscopy. These experiments have been repeated at least for three independent times with similar results.

---

modern *Y. pestis* isolates and *Y. pseudotuberculosis*[53], encode exactly identical YspE1. However, homologs of *yspE1-yspE2* are extremely divergent across the different *Y. pseudotuberculosis* strains (Fig. 4a, b) and often contain only one or two homologs genes or encode truncated proteins lacking NEL domain. The fact that *Y. pseudotuberculosis yspE1* homologs are highly diversified in sequences across the different

strains provoked our interest to exam whether *Y. pseudotuberculosis* is able to degrade GBPs as well. The cell transfection assay results showed unambiguously that YPIII, IP2666pIB1 and PB1 + YspE1 homologs, representing the three major clusters of YspE1 homologs in various *Y. pseudotuberculosis*, cannot degrade any of hGBPs. Domain-swapping experiments results found that their NEL domains are

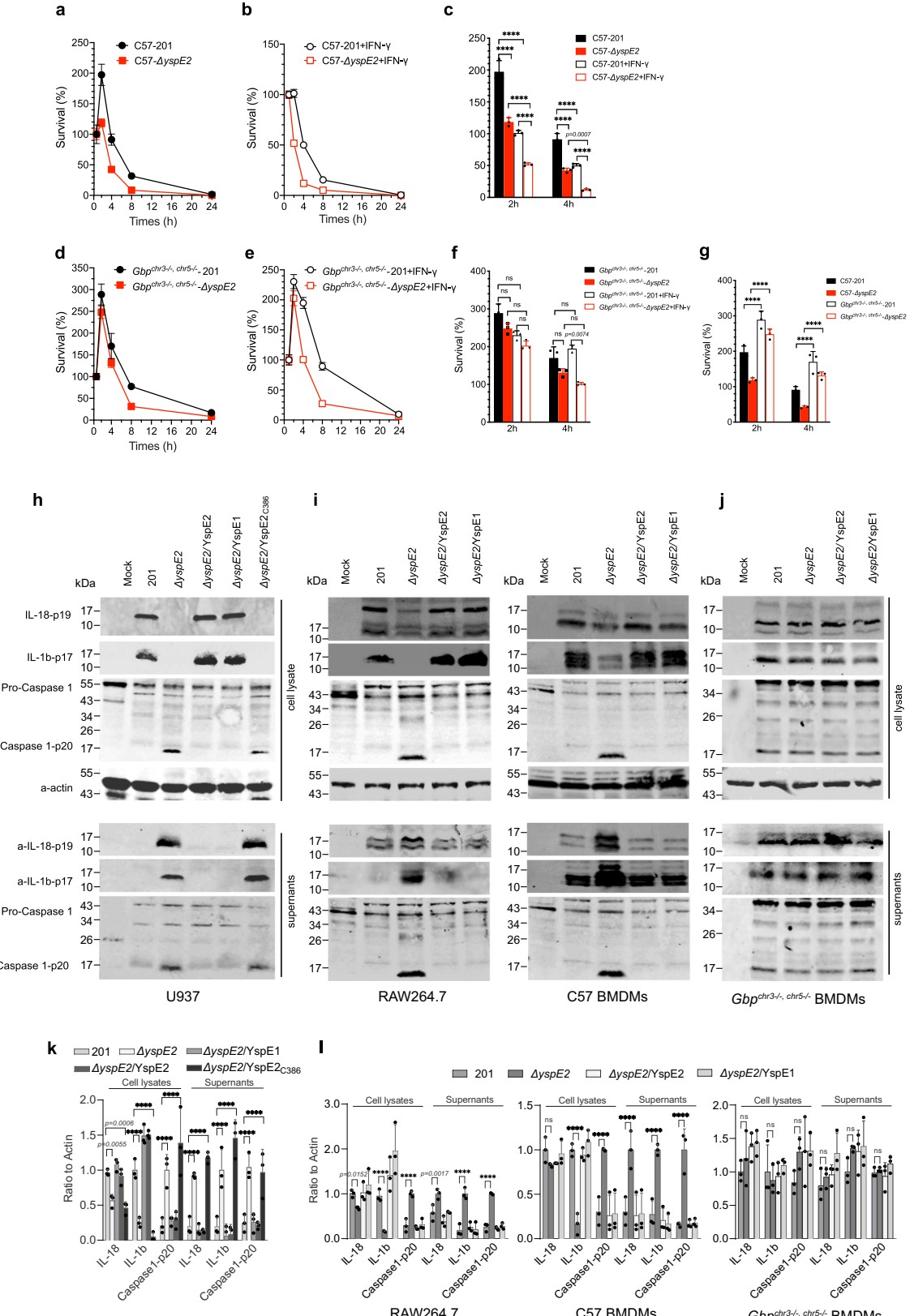

catalytic active and the sequence variations in the LRR domain lead to the failure of hGBP1 degradation (Fig. 4e–g), in line with the well-established knowledge that LRR domain is responsible for the substrates recognition by the NELs. The extreme diversity of *yspE1-yspE2* loci in *Y. pseudotuberculosis* implies that it is not important or essential for this enteropathogen. Notably, *yspE1-yspE2* loci in the *Y. pestis* 0.PE

(Pestoides) strains, which have just diverged from *Y. pseudotuberculosis*, appears to be unstable since 0.PE4 strains harbor this loci while 0.PE2 and 0.PE3 strains do not, but thereafter stabilized in modern *Y. pestis* isolates (Fig. 4b, Supplementary Table 2). We hypothesize that complete *yspE1-yspE2* loci could be formed gradually, probably by gene duplication, during evolution from *Y. pseudotuberculosis*, and

**Fig. 7 | YspE2 promotes the survival of *Y. pestis* in macrophage and inhibits the inflammasome activation in an E3 ligase activity dependent manner.** The percentage of intracellular survival of *Y. pestis* 201 and *ΔyspE2* in BMDMs isolated form C57BL/6 J (**a**, **b**) or *Gbp^chr3−/−, chr5−/−* mice (**d**, **e**). BMDMs were directly infected with the *Y. pestis* 201 or *ΔyspE2* (**a**, **d**), or primed with IFN-γ prior to infection (**b**, **e**). The living bacteria numbers inside the infected BMDMs were measured using a gentamycin protection assay. Triplicates samples were analyzed for each experiment and the averages with standard deviations were shown. Survival percentages of *Y. pestis* 201 and *ΔyspE2* in the wild type BMDMs (**c**) and *Gbp^chr3−/−, chr5−/−* BMDMs (**f**) at 2 and 4 hpi were shown, and comparisons between survival percentages of *Y. pestis* strains in the wild type BMDMs and *Gbp^chr3−/−, chr5−/−* BMDMs were shown in **g**. Two-way ANOVA

with Bonferroni's multiple comparisons test was performed to analyze the significance of difference in the survival percentages. **h** U937 cells were infected with *Y. pestis* 201, *ΔyspE2*, *ΔyspE2*/YspE1, *ΔyspE2*/YspE2 or *ΔyspE2*/YspE2_C386A strains as indicated, and both the culture medium and cell lysates were analyzed for IL-1β, IL-18 and caspase 1-p20. Inhibition of inflammasome activation in human macrophage-like U937 cells by *Y. pestis* requires YspE1/YspE2 E3 ligase activity. **i** RAW264.7 cells and BMDMs isolated from C57BL/6 J or *Gbp^chr3−/−, chr5−/−* mice (**j**) were infected as described in **h**. **k, l** Represented the quantitative and statistical analysis results of **h** and *P*, respectively, from three independent experiments. Two-way ANOVA was performed to analyze the significance of difference in protein levels (*\*p < 0.05; \*\*p < 0.01; \*\*\*p < 0.001; \*\*\*\*p < 0.0001*).

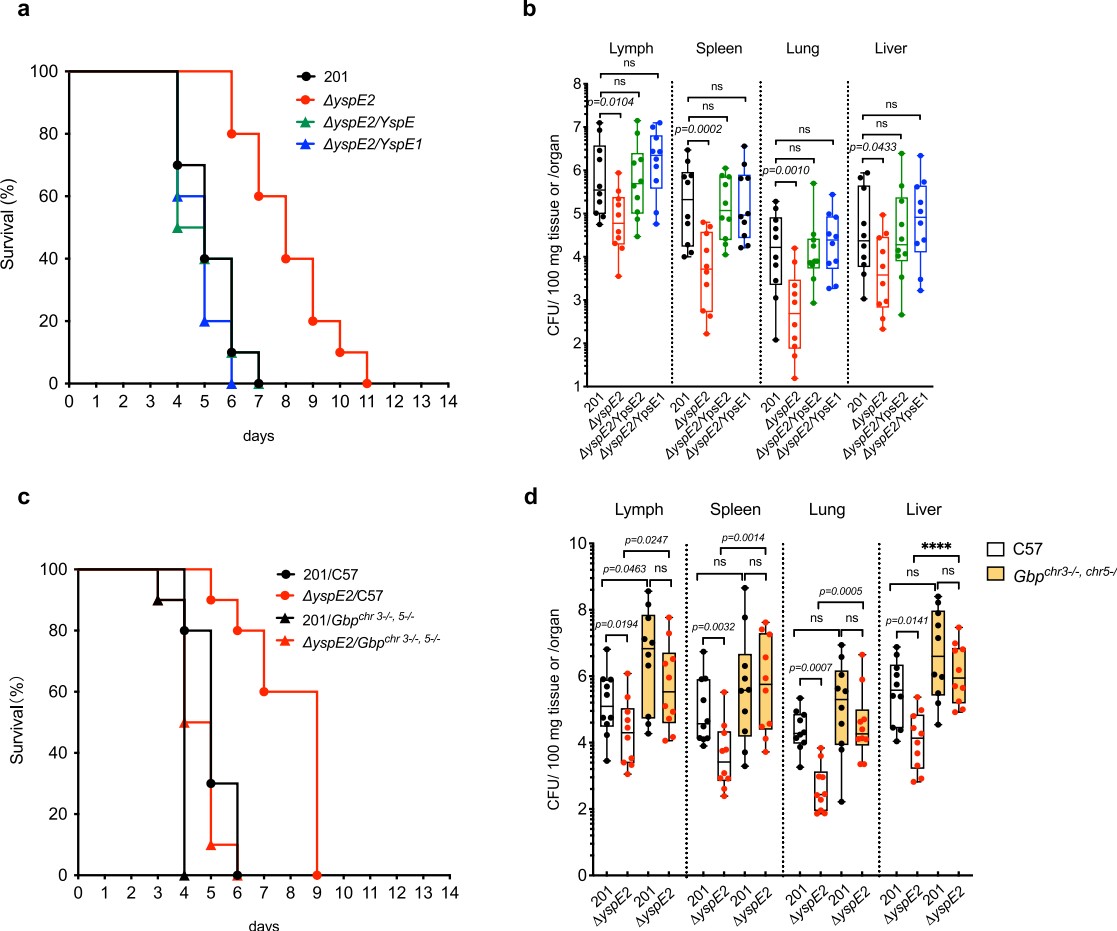

**Fig. 8 | YspE1/YspE2 contribute significantly to the virulence of *Y. pestis* and *Gbp^chr3−/−, chr5−/−* mice were more susceptible to *Y. pestis* infection. a** Groups of female C57BL/6 J mice (*n* = 10 per group, 6 to 8 weeks) were *s.c.* challenged with *Y. pestis* 201 (WT), *ΔyspE2*, *ΔyspE2*/YspE1 or *ΔyspE2*/YspE2, and survival curves were plotted. *Δ yspE2* significantly attenuated in virulence in mice (*p* < 0.001) and no significant difference exist between survival curves of 201 and YspE1- or YspE2-complemented *yspE2* mutant. Log-rank test was performed using GraphPad Prism 8.0 to analyze the significance of difference in survival curves. **b** Bacterial loads of *ΔyspE2* in tissues of mice harvested at 4 dpi were significantly lower than those of *Y. pestis* 201 in *s.c.* challenged C57BL/6 J mice. One-way ANOVA was performed using GraphPad Prism 8.0 to analyze the significance of difference in bacterial loads in the different tissues. **c** Groups of C57BL/6 J mice and *Gbp^chr3−/−, chr5−/−* mice (*n* = 10 per group, 6 to 8 weeks) were *s.c.* challenged with *Y. pestis* 201 or *ΔyspE2*. Compared to C57BL/6 J mice, *Gbp^chr3−/−, chr5−/−* mice are much more susceptible to both strains in subcutaneous infection (*p* < 0.001), and the significance of difference between survival curves of the two strains in C57BL/6 J mice (*p* < 0.001) were greatly

decreased in *Gbp^chr3−/−, chr5−/−* mice but still significant (*p* = 0.037). The differences between the numbers of *Y. pestis* 201 and *ΔyspE2* in tissues disappeared in *Gbp^chr3−/−, chr5−/−* mice (**d**). Mice were infected as described in **c** and the tissues were harvested when the mice were moribund, i. e. 4 dpi. for C57BL/6 J mice and 3 dpi. for *Gbp^chr3−/−, chr5−/−* mice. Results are shown in box and whiskers plot with bottom and top of the box are the 25th and 75th percentile and lines in box are medians. The upper and lower ends of whiskers represent the maxima and minima of data points, respectively. The limit of detection for each tissue are as follows: 21 CFU/100 mg lung tissues, 40 CFU/spleen, 80 CFU/groin lymph node, 19 CFU/100 mg liver tissues. Two-way ANOVA was performed to analyze the significance of difference in bacterial loads in the different tissues (*\*p < 0.05; \*\*p < 0.01; \*\*\*p < 0.001; \*\*\*\*p < 0.0001; ns, not significant*). All the experiments for survival curves have been performed for 3 independent times and experiments for bacterial loads have been performed for 2 independent times and similar results have been acquired, and only one representative result was shown.

harboring functional YspE1/YspE2 E3 ligases yields survival benefit in host or fitness to the new flea-borne transmission lifestyle of the newly formed specie of *Y. pestis*, whereby they were subsequently stabilized owing to the competitive advantages. Obviously, this hypothesis needs further validation because we analyzed *yspE1-yspE2* gene locus only in the currently available completed genomes of *Y. pestis* and *Y. pseudotuberculosis*.

Our results reveal one of the unique virulence mechanisms acquired by *Y. pestis* genome that has underwent a significant reductive evolution, in addition to the two *Y. pestis*-specific plasmids pPCP1 and pMT1. Notably, three representative YspE1 homologs in *Y. pseudotuberculosis* YPIII, IP2666pIB1 and PB1+ have higher sequence identity to YspE1 than *Shigella* IpaH 9.8, while none of them can degrade GBPs as IpaH9.8 does, even though IpaH9.8 is different from YspE1 in substrate preferences to hGBPs and mGBPs. We suppose that some crucial structural confirmation for binding to hGBPs might be present in YspE1/YspE2 and IpaH9.8 but not in their *Y. pseudotuberculosis* homologs. Bacterial pathogens including *Y. pestis*, as well as *Shigella, Francisella, Legionella* and *Salmonella* have evolved multifarious virulence factors to target different hGBPs, implying that the GBPs-mediated cell-autonomous defense plays vital roles in controlling infection of both obligate and facultative intracellular pathogens.

# Methods

## Plasmids and bacterial strains

The plasmids and bacterial strains used in this study were listed in Supplementary Table 1. *Y. pestis* 201 strain belongs to the biovar Microtus and it is highly virulent to mice but avirulent to humans[54,55]. *Y. pestis* strain 201 and *Y. pseudotuberculosis* strain YPIII and Pa3606 were cultured in Luria-Bertani (LB) broth at 26 °C or chemically defined TMH medium[56], with or without 2.5 mM calcium at 26 °C or 37 °C, as indicated for specific experiments. *E. coli* DH5 α or BL21(DE3) were grown in LB broth. When necessary, antibiotics were added to the medium at the following concentrations: 100 mg/ml ampicillin, 20 mg/ml chloramphenicol, 40 mg/ml gentamicin, and 50 mg/ml kanamycin.

## Antibodies and reagents

Caspase-1 p20 (#PA5-94989), IL-18 (#PA5-79479) and IL-1β (#701304) antibodies were obtained from Thermo Fisher Scientific (Waltham, MA, USA). Antibodies for Ubiquitin linkage-specific K48 (ab140601), Ubiquitin linkage-specific K63 (ab179434), hGBP1 (ab121039), hGBP2 (ab203238), hGBP3 (ab74061), hGBP4 (ab173697), hGBP5 (ab130569), hGBP6 (ab125704), hGBP7 (ab104293) were from Abcam (Cambridge, UK). Antibodies for β-actin (RM2001), c-Myc Tag (KM8003), Flag-Tag (KM8002) were obtained from Sungene Biotech (Tianjin, China). Ubiquitin (A-100) antibody was from Boston Biochem (Cambridge, MA, USA). IRDye 800CW-conjugated goat anti-rabbit antibody (C90529-19) and IRDye 800CW-conjugated goat anti-mouse antibody (C81106-03) were from LI-COR Biosciences (Lincoln, NE, USA). YspE1, YspE2 and YopM antibodies were polyclonal antibodies raised in rabbit using the corresponding purified recombinant full-length proteins and prepared by our laboratory as described previously[57].

Pierce BCA Protein Assay Kit (#23227), glass coverslips in 15 mm plates and Lab-Tek chambered coverglass (155411) were Thermo Fisher Scientific products. Ubiquitin human (U5382), mouse and human IFN-γ, MG132 (M8699), ANTI-FLAG® M2 Affinity Gel (A2220), Anti-c-Myc Affinity Gel (E6654), calmodulin (C4874), Thrombin (T6634) and ATP were from Sigma-Aldrich (St Louis, MO, USA). Cytochalasin D (1233) and cAMP Elisa kit (KGE002B) were from R&D Systems (Emeryville, CA, USA). UbcH5a/UBE2D1 (#E2-616), Ubiquitin Conjugation Initiation Kit (K-995), K48 (UM-K480), K63-linked ubiquitin chains (UM-K630) and Recombinant Human UbcH (E2) Enzyme Set (K-980) were Boston Biochem products. X-treme GENE HP DNA Transfection Reagent (42980400) were from Roche (Germany). Glutathione affinity gel (17-0756-01) and PD-10 desalting columns was obtained from GE

Healthcare (Pittsburgh, PA. USA). Ni-NTA agarose (30210) was obtained from Qiagen (Valencia, CA, USA). The Amicon Ultra-15 regenerated cellulose centrifugal filter device (3,000-Da molecular weight cutoff) and the Immobilon-P transfer membrane were purchased from Millipore (Bedford, MA, USA).

## Cell cultures

HEK293T, HeLa and RAW264.7 were maintained in Dulbecco's modified Eagle's medium (DMEM) and U937 was maintained in RPMI-1640 (HyClone, Little Chalfont, UK) containing 10% fetal bovine serum (Gibico, Grand Island, NY, USA) at 37 °C in a 5% $CO_2$ incubator).

## Construction of the *Y. pestis* mutants and the complemented strains

The coding sequences of *yspE1* and *yspE2* were eliminated via homologous recombination using suicide vector pDS132. The up and downstream flanking regions of the target genes were amplified by PCR from genomic DNA of *Y. pestis* 201. Mutant cassettes, which were consisted of ~300 bp homologous sequences flanking the mutated genes at both sides of DNA fragments, were generated by overlap extension-PCR and cloned into pDS132. The recombinant plasmids were transformed into *E. coli* S17λpir. Then, the *yspE1* and *yspE2* deletion were introduced into the *Y. pestis* 201 strain by conjugation and allelic exchange. The *yp_3417* mutant was generated by replacing the coding sequences of *yp_3417* with the kanamycin resistance cassette using the λRed recombination system as described previously[58].

Plasmids pBAD24 and pACYC184 were used to construct complemented strains. PCR-generated DNA fragments containing the *yspE1, yp_3417* or *yspE2* coding sequences were cloned into pBAD24 and the expression of the cloned genes were induced by addition of 0.2% arabinose. PCR-generated DNA fragments containing the coding sequences of these genes and the 300 bp-upstream sequence to the interested gene were cloned into pACYC184. The recombinant plasmids were individually introduced into the corresponding mutants, yielding the complemented mutant strains.

## Purification of recombinant proteins

*E. coli* BL21 (DE3) strains containing plasmids pGEX-4T-2-YspE1, pGEX-4T-2-Yp_3417, pGEX-4T-2-YspE2, pGEX-4T-2- YspE1$_{C407A}$, pGEX-4T-2-YspE2$_{C386A}$, pGEX-4T-2-YspE1$_{LRR}$, pGEX-4T-2- YspE2$_{LRR}$ and pSUMO-hGBP1 were grown in LB medium with appropriate antibiotics. When $OD_{600nm}$ of the cultures reached 0.6, protein expressions were inducted with 1 mM isopropyl-B-Dthiogalactopyranoside (IPTG) overnight at 20 °C in an incubator at 100 rpm. His-tagged proteins were purified by affinity chromatography using Ni-NTA beads. GST-tagged proteins were purified by glutathione sepharose beads 4B as described previously[35]. The eluted solutions containing the purified proteins were exchanged to phosphate-buffered saline (PBS) using PD-10 desalting columns. Finally, the purified His or GST tagged proteins were excised by ULP1 (Ubl-specific protease 1) and Thrombin, respectively.

## Multiple sequences alignment analysis

We downloaded 38 complete genomes of *Y. pestis* and 23 complete genomes of *Y. pseudotuberculosis* available from NCBI GenBank database on October 10th, 2020. The DNA sequences of gene *yspE1*, *yp_3417*, *yspE2* from *Y. pesti*s 91001 (Accession: GCA_000007885.1) were used as reference to blast against all the complete genomes and then the corresponding sequences of three genes in each genome were extracted. The extracted sequences having less than 80% of the corresponding total length of the reference gene were excluded from further analysis. The software MEGA (v7.0.26) was used to do multiple sequences alignment and compute pairwise distances. We used (1 − the value of pairwise distance) ×100 as the identity between two sequences and displayed the result using the package of heatmap in R (v3.6.1).

## Surface plasmon resonance (SPR) analysis

Real-time monitored SPR assays were performed to determine the bindings of the purified YspE1, YP_3417, YspE2, YspE1$_{LRR}$ and YspE2$_{LRR}$ to hGBP1 or hGBP4 as indicated using a Biacore T200 instrument. $Y.$ $pestis$ E3 ligase proteins or their LRR domains were diluted with 10 mM acetate buffer (pH 4.0) to a concentration of 20 µg/ml and coupled to a NHS/EDC-activated Carboxylated dextran chips (CM-5 sensor chip) to yield approximate 100 resonance units (RU). Subsequently, hGBP1 (0–20 µM) in HBS buffer (20 mM pH = 7.4 HEPEs, 150 mM NaCl, 3.4 mM EDTA and 0.005% (v/v) P20) was injected for 180 seconds at a flow rate of 30 µL/min and the protein-protein interactions were monitored in real time. Regeneration of the sensor chip was achieved with a 30-second wash with 2.5 mM glycine-HCl at a flow rate of 30 µL/min. All measurements were performed at 25 °C and the data was analyzed using the Biacore T200 Evaluation Software 2.0.

## In vitro ubiquitination assays

For the in vitro ubiquitination assay, a reaction mixture containing 10 µg purified proteins (YspE1, YP_3417, YspE2, YspE1$_{C407A}$ or YspE2$_{C386A}$), 2 µg FLAG-ubiquitin, 25 µM UbcH5b, 1 µL 20×E1, 2 µL 10×Mg$^{2+}$-ATP and 5 µL 10×reaction buffer was incubated at 37 °C for 90 minutes[59]. Reactions were quenched with SDS loading buffer, boiled for 10 mins and then separated on a 4–20% SDS-PAGE. Then, the proteins on the gel were transferred onto an immobilon-P transfer membrane and subjected to immunoblot analysis with anti-FLAG mouse monoclonal antibody and IRDye 800CW-conjugated goat anti-mouse IgG. The images of the immunoblotting results were obtained using the Odyssey SA imaging system (LI-COR).

## Secretion analysis of YspE1 and YspE2 by $Y.$ $pestis$

$Y.$ $pestis$ 201 was grown at 26 °C in TMH medium with or without calcium to OD$_{620nm}$ of 1.0, then shifted to 37 °C for 2 h's incubation to induce T3SS. The supernatants containing the secreted proteins were collected by centrifugation and concentrated using an Ultra-15 centrifugal filter device. The concentrations of proteins were determined with BCA protein assay kit. The bacterial pellets were resuspended in PBS. Equal amounts of proteins in supernatants and pellets were boiled in SDS loading buffer, separated by 12% SDS-PAGE and analyzed by immunoblotting with anti- YspE1, anti- YspE2 and anti-YopM rabbit polyclonal antibodies.

## Survival capability of $Y.$ $pestis$ strains in BMDMs

BMDMs were isolated from C57/BL/6 J mice and differentiate for 7 day in presence of recombinant mouse M-CSF (20 ng/ml) as described previously[60]. For the following experiments, BMDMs were harvested using a cell scraper and seeded into 24-well plates at a density of $5 \times 10^5$ cells/well the day before infection. BMDMs were infected with $Y.$ $pestis$ strain 201 or $\Delta yspE2$ at an MOI of 20, and the plates were centrifuged to promote the bacteria-cell contact. At 0.5 hpi, gentamicin at 30 µg/ml was added to each well to kill the extracellular bacteria. After incubation for additional 0.5 h, the culture medium was replaced with fresh DMEM containing 10% FBS and 2 µg/ml gentamicin, which was sustained till the end of the experiment. At 0.5, 2, 4 and 8 hpi., the culture medium was removed and the BMDMs were thoroughly washed with PBS and lysed in sterile H$_2$O containing 0.1% Triton X-100 for 15 min at room temperature. The number of living bacteria was determined by plating the diluted bacterial suspensions onto a Hottinger's agar plate in triplicate. This experiment was independently performed three times in biological triplicates and survival curves were plotted using GraphPad Prism 8.0, which were shown as the mean ± standard deviation ($n = 3$).

## Detection of hGBPs degradation by western blot

HeLa cells stably expressing hGBPs and RAW264.7 were seeded in 6-well plates and grown to reach 70–80% confluency. Cells were primed with human or mouse IFN-γ for 16 h to stimulate the expression of endogenous GBPs. $Y.$ $pestis$ 201 and the various mutants of $Y.$ $pestis$ were grown to OD$_{620nm}$ 1.0 at 26 °C, followed by 2 h incubation at 37 °C. Cells were then infected with the individual strains (MOI = 20) for the indicated periods. After infections, cells were washed twice with PBS and lysed in lysis buffer containing 25 mM Tris-HCl (pH 7.6), 150 mM NaCl, 0.5% Triton X-100 and the protease inhibitor mixture. Then, the samples were separated by 12% SDS-PAGE and immuno-blotted with anti-GBPs (hGBP1-hGBP7) antibodies. For plasmids transfection, HeLa cells stably expressing GFP-tagged hGBPs were individually transfected with p3×Flag-CMV-YspE1, p3×Flag-CMV-YP_3417, p3×Flag-CMV-YspE2, p3×Flag-CMV-YspE1$_{C407A}$ and p3×Flag-CMV- YspE2$_{C386A}$ using X-tremeGENE HP DNA transfection reagent. After 36 h of transfection, cells were lysed and subjected to immuno-blotting analysis as described above.

## Confocal microscopic analysis of hGBPs during infection

Bacterial strains containing pUC19-Scarlet were grown at 26 °C to OD$_{620nm}$ 1.0 prior to infection. HeLa cells stably expressing the GFP-hGBPs were seeded onto glass coverslips in 15 mm dishes and primed with human IFN-γ 16 h prior to infection to stimulate the expression of endogenous GBPs. Cells were individually infected with the above strains at an MOI of 10 for the indicated time periods. Then, the infected cells were washed thoroughly, fixed with 4% paraformaldehyde and fluorescence images were acquired using an Andor Dragonfly spinning disk confocal microscope. For real-time laser confocal scanning experiments, cells were seeded in Lab-Tek chambered cover-glass and infected with the individual bacterial strains at an MOI of 20. During infection, images were taken every 5 mins until 8 hours of infection, and 5–10 fields of view were chosen for each sample using spinning disk system with Nikon ECLIPSE Ti-E microscopy.

## Detection of protein interactions and ubiquitination in vivo

For immunoprecipitation, HEK293T cells in 6-well plates at 70-80% confluency were co-transfected with pCMV-Myc-hGBPs and p3×Flag-CMV plasmids expressing YspE1, YP_3417, YspE2 and their mutants. Cell transfection and lysis were performed as described above. The cell lysates with 3×Flag tagged proteins were incubated with anti-Flag M2 beads overnight at room temperature. Then the beads were washed three times with the lysis buffer, and the immunoprecipitates on the beads were separated by SDS-PAGE and subjected to immunoblotting detection using anti-cMyc and anti-FLAG antibodies. For in vivo ubiquitination assay, the experimental procedures were similar to the immunoprecipitation assay described above except that the lysates containing Myc-tagged proteins were incubated with the Myc beads, then ubiquitination of the proteins were detected by anti-Ub, -Ub-K48 and -Ub-K63 antibodies.

## Inflammasome activation assays

HeLa, RAW264.7, U937 cells or BMDMs isolated from 8-10 weeks old C57BL/6 J or $Gbp^{chr3,5-/}$ mice were grown to about 90% confluency before infection. Phorbol 12-myristate 13-acetate (PMA, 100 nM) was added for U937 cells to differentiate them into macrophages. HeLa cells were infected with $Y.$ $pestis$ 201 strain, $\Delta yspE1$, $\Delta yspE2$-pACYC184-$yspE2$ or $\Delta yspE2$-pACYC184-$yspE1$ at an MOI of 100, and the supernatants and cell lysate were collected separately at 2 hpi. BMDMs, RAW264.7 and U937 cells were infected with above strains at an MOI of 20. After 30 min, the culture medium was replaced with fresh medium containing gentamicin to inhibit the overgrowth of bacteria, and the supernatants and cell lysate were collected at 4 hpi. Proteins in the supernatants were TCA precipitated and the cell lysates were processed directly with SDS loading buffer, and the samples were then analyzed by immunoblotting using anti-caspase-1, anti-IL-1 β and anti-IL-18 antibodies[61].

## Adenylate cyclase activity assay

The plasmids pBAD24-YspE1$_{100-CyaA}$, pBAD24-YP_3417$_{100-CyaA}$, pBAD24-YspE2$_{100-CyaA}$, 201-pBAD24-YopM100-CyaA, and 201-pBAD24-CyaA were constructed to express the Cya hybrid proteins[62]. The wild type 201 strain and 201 strains containing above recombinant plasmids were cultured with 0.02% arabinose to OD$_{620nm}$ = 1.0, then shifted to 37 °C for an additional 2 h to fully induce T3SS. HeLa cells were seeded in 12-well plate and allowed to grown until 70-80% confluency. At 30 minutes prior to infection, cytochalasin D were added at a final concentration of 5 μg/ml. Cells were infected with the various strains for 2 h at 37 °C (MOI = 100). After infection, cells were washed three times with PBS and lysed in denaturing conditions (HCl 50 mM, Triton 0.1%, 100 °C for 5 min). The cAMP concentrations were detected by cAMP ELISA kit. To analyze adenylate cyclase activity in HeLa cells, the extracts of HeLa cells were cleared by centrifugation at 7500 g and the supernatants were supplemented with 2 mM ATP and 0.1 μM calmodulin, followed by incubation for 10 min at 30 °C. Results were expressed in adenylate cyclase activity units per $10^8$ bacteria[63].

## Animal experiments

C57 BL/6 J mice were purchased from Vitalriver Company (Beijing, China). The animals were maintained in an animal facility at ambient temperature of 20-25 °C and relative humidity of 30–60%. Animals were handled in strict accordance to the Guidelines for the Welfare and Ethics of Laboratory Animals of China and all the animal experiments were approved by the Institutional Animal Care Committee of Beijing Institute of Epidemiology and Microbiology (protocol number IACUC-DWZX-2020-062). *Y. pestis* 201 strain, *ΔyspE2*, *ΔyspE2*-pACYC184- *yspE1* and *ΔyspE2*-pACYC184-*yspE1* were grown in LB until OD$_{620nm}$ of ~1.0, and bacterial cells were collected by centrifugation and the bacterial suspensions in PBS were prepared. Groups (*n* = 10) of 6 to 8 week-old female mice (C57BL/6 J or *Gbp*$^{chr3-/-,chr5-/-}$) were *s.c.* challenged with 100 μl of bacteria suspension (about 100 CFU) per mouse and the challenged mice were observed once daily continuously for 14 days. The time of death was recorded and the survival curves for each of mouse group were plotted using GraphPad Prism 8.0. To determine the bacterial burden in organs, livers, spleens and lungs of the challenged mice were harvested when the mice were moribund, i.e. 4 dpi. for C57BL/6 J mice and 3 dpi. for *Gbp*$^{chr3-/-,chr5-/-}$ mice, due to the more susceptibility of *Gbp*$^{chr3-/-,chr5-/-}$ mice to *Y. pestis* infection., and the collected tissues were homogenized in sterile PBS for measurement of living bacteria as described previously[64].

## Generation of *Gbp*$^{chr3-/-,chr5-/-}$ mice

To generate of *Gbp*$^{chr5-/-}$ mice, two targeting gRNAs (5'- gcagcttctc-cagggaatca −3' & 5'- gctctggtttagtgggggtg −3') and in-vitro translated Cas9 mRNA were co-microinjected into C57BL/6 zygotes. Deletion of the targeted *Gbp* locus on mouse chromosome 5 was screened by PCR and validated by DNA sequencing. The following four primers were used to detect the *Gbp*$^{chr5}$-locus deletion: Wild type-F- ggccaacgcat-gacaaagaa; Wild type-R-ggtatgtcacattatttctggtggc; *Gbp*$^{chr5-/-}$ KO-F-agatccttgctctacttgaagttt; *Gbp*$^{chr5-/-}$ KO-R- gcctctctccctaccgattcttc. Then, *Gbp*$^{chr5-/-}$ mice were crossed with *Gbp*$^{chr3-/-}$ mice (*Gbp*$^{chr3-/-}$-Type-II)[31] to generate *Gbp*$^{chr3-/-,chr5-/-}$ mice.

## Reporting summary

Further information on research design is available in the Nature Research Reporting Summary linked to this article.

## Data availability

All data generated or analyzed during this study are included in this published article and supplementary information files. Source data are provided with this paper.

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

## Acknowledgements

We thank the Public Instrument Platform of School of Life Sciences at Peking University and National Center for providing experimental instrument support. We thank Kai Wang at Biomedical Analysis Academy (Beijing, China) for help in confocal microscope analysis. This work was supported by the National Natural Science Foundation of China (Nos. 81902032 to S.C., 32070136 to Z.D., and 31470242 to R.Y.).

## Author contributions

Z.D. and R.Y. designed the study. S.C. performed the experiments unless otherwise noted. Y.J., Y.R., Y.Y., X.G., Y.T., H.C., G.W., Y.Zhang, T.W. and Y.Zhou. performed the mouse experiments. Y.W., Z.D., S.Q., Y.C. and Y.S. analyzed the sequence data of different *Y. pestis* and *Y. pseudotuberculosis* genomes. F.S. and W.J. provided knockout mice. F.S. reviewed the manuscript and raised constructive suggestions. Z.D., S.C. and R.Y. wrote the manuscript.

## Competing interests

The authors declare no competing interests.
