## [Peer Review File · Nature Communications]

Yersinia pestis E3 ligases ubiquitinate and degrade GBPs to antagonize the cell-autonomous defense, an arsenal acquired during evolutionREVIEWER COMMENTS

Reviewer #1 (Remarks to the Author):

Previous studies have shown that quarylate-binding proteins (GBPs) have been shown to recognize pathogen-containing vacuoles, including the Yersinia-containing vacuole (YCV), leading to both lysis of this intracellular niche and induction of the inflammasome. However, the contributions of GBPs during in vivo infection with *Y. pestis*, the bacterium that causes the human disease known as the plague, has not been previously described. Here, Cao et al. present data suggesting that two newly identified genes in *Y. pestis* 201 (YP_3416 and YP_3418) help *Y. pestis* to evade recognition by GBPs and the innate immune system to promote host colonization. Both YP_3416 and YP_3418 have N-terminal LRR domains (a protein interaction domain) and C-terminal NEL domains (a Novel E3-ubiquitin Ligase domain). Using a combination of biochemical, cell culture, and animal infections, the authors present data supporting that: 1) YP_3416 and YP3418 are secreted into host cells via the Ysc type 3 secretion system, 2) infection of host cells by *Y. pestis* 201 results in ubiquitination and degradation of host quarylate-binding proteins (GBPs), 3) ubiquitination and degradation of host GBPs is dependent on YP_3416 and YP_3418, 4) GBPs inhibit intracellular survival of *Y. pestis* 201, 5) YP_3416 and YP_3418 inhibit GBPs recognition of the Yersinia-containing vacuole, 6) YP_3416 and YP_3418 contribute to virulence during bubonic plague, which is dispensable in the absence of mouse GBPs. The manuscript is clearly written, and provide several independent lines of evidence supporting the role of these proteins in *Y. pestis* virulence. I think these data represent several exciting discoveries that merit consideration for publication. First, YP_3416 and YP_3418 are chromosomally encoded, and to my knowledge represent the first effectors secreted by the Ycs T3SS that are not encoded on the pCD1 plasmid (which encodes the T3SS and the seven bonafide T3SS-dependent effectors secreted by *Y. pestis* known as the Yops). Second, these two genes are conserved in the highly virulent *Y. pestis* strains, but appear to be less conserved in *Y. pseudotuberculosis*, suggesting that acquisition of YP_3416 and YP_3418 may have contributed to the transition of *Y. pestis* from a gastrointestinal pathogen to a blood-borne pathogen. Finally, the authors also present the first data indicating that GBPs contribute to resistance to *Y. pestis* infection, supporting the hypothesis that a transitional intracellular niche for *Y. pestis* during bubonic plague contributes to overall virulence of this facultative intracellular bacterium. That being said, the authors' conclusions would be greatly strengthened by the addition of replicate data for many of the figures and better description of statistics during animal studies.

Comments:

Figure 1: I appreciate using both in vitro and cell culture models to monitor T3SS-dependent translocation of YP_3416 and YP_3418. However, I am confused why some bands are highlighted as non-specific and marked with "*" when they are absent in the individual gene deletions. This would indicate to me they are not non-specific. Also, at 84% protein identity, how do you not have cross reactivity between 3416 and 3418? The methods do not describe if these are against full length proteins are only peptides that specific for each protein.

Figure 2: All of these data in this figure is from one representative experiment. The rigor of these data would be significantly improved if you included quantification of bands relative to actin for multiple experiments, and include the statistics to support your observations.

Figure 3 and Lines 130-140: In this section, you argue that *Y. pseudotuberculosis* YPIII is not able degrade GBPs despite of the fact that it encodes homologs to YP_3416 and YP_3418. However, in Line 137 you state that this strain of YPIII lacks the pYV plasmid that encodes the T3SS. Why are you using a T3SS deficient strain for this comparison? Wouldn't a pYV positive strain be much more appropriate, and the only way to specifically show that the homolog is *Y. pseudotuberculosis* are not active. You should also need data from replicates for 3A and 3B.

Figure 4: All of these western blot data in this figure is from one representative experiment. The rigor of these data would be significantly improved if you included quantification of bands relative to actin for multiple experiments, and include the statistics to support your observations.

Figure 5: I am a little concerned by the interpretation of this figure. I am not sure you can conclude that GBP1 co-localizes with *Y. pestis*. To conclude this, I think you need to generate a Pearson correlation coefficient in order to have some statistical analysis to back this up. It appears that hGBB1 is everywhere in the cell and the reason there is not a colocalized signal is because it is either not as well expressed or degraded during *Y. pestis* infection (latter is more likely based on your other data). Moreover, bacteria were grown under conditions that should induce the T3SS and inhibit phagocytosis. This would suggest that the majority of bacteria should be extracellular. Does this mean that the GBPs are being recruited to the plasma membrane under attached bacteria? If YPIII is lacking pYV as described in Line 137, then is there significantly more intracellular bacteria than in the *Y. pestis* 201 infected situation? Again, does the lack of the T3SS artificially result in a lack of degradation of GBP?

Figure 6: If the degradation of GBPs is the only function of YP_3416 and YP_3418 then wouldn't you expect that the phenotypes would overlap in D and E? This was not clearly addressed in the manuscript. The rigor of the data in G, H, and I would be significantly improved if you included quantification of bands relative to actin for multiple experiments, and include the statistics to support your observations. The observed decrease in intracellular survival as compared to previous papers that the authors discussed in Lines 297-305 is likely due to the extremely high dose of, and extended incubation period with, gentamicin that was employed in these assays (see PMID: 29312891). Repeating these assays with a lower amount of antibiotic may further differentiate a GBP/ YP_3418 phenotype in the macrophages.

Figure 7: Survival curves lack statistical analysis. Are these curves significantly different? The most likely tissue that the bacteria will be intracellular are the lymph nodes. Also, the lymph nodes are a bottle neck for subsequent dissemination, but colonization of these tissues are missing. Therefore, a kinetic time course analysis of lymph node colonization in WT and KO mice would greatly increase the authors' hypothesis that YP_3418 contributes to intracellular survival that improves virulence. Legend is missing group size, sex of mice, and number of times the experiment was repeated (should be repeated to show reproducibility).

Minor Comments:

Line 174: Where is binding data for 1, 2, 5, and 6?

Line 191: Adding the KD for IpaH9.8 would be helpful to provide context.

Reviewer #2 (Remarks to the Author):

The authors report the characterization of YspE1 and YspE2, two novel ubiquitin E3 ligases of the NEL class in *Yersinia*, which the authors demonstrate degrade several GBPs. The role of YspEs therefore appears similar to the related NEL ligase IpaH9.8 in *Shigella*, which is known to also antagonize GBP function through ubiquitylation (Li Nature 2017; Wandel, Cell Host Microb 2017). Similar to IpaH9.8, YspE2 also forms K48 linked chains on GBP1 that result in proteasomal degradation of GBPs. Deletion of YspE1s caused enhanced GBP association with *Yersinia*, improved anti-bacterial activity of host cells in vitro and reduced mortality in infected mice, again similar to the situation in *Shigella* (Wandel Nat Immunol 2020).

Taken together, while the report of two novel NEL class E3 ligases in *Yersinia* and the identification of GBPs as their substrate is interesting, particularly to microbiologist in the *Yersinia* field, in more general terms the manuscript merely confirms the important role of GBPs in anti-bacterial immunity and that bacteria overcome GBP-mediated defence through ubiquitin-dependent proteolysis by providing another example in a different pathogenic bacterium.

Major points:

1. Line 210 – Using epithelial cells, the authors report that “...GBP1 was largely co-localized with” Yersinia. The authors should determine the fraction of GBP1-coated bacteria, as well as the fraction of cytosol exposed bacteria. Is the majority of Yersinia accessing the cytosol and are GBPs directly recruited to bacteria? Or are GBPs accumulating on Yersinia containing vacuoles? Higher resolution images as well as quantification of marker colocalization will be needed. Galectin costaining might be useful.
2. The degradation of endogenous GBPs (or at least selected GBPs) upon Yersinia infection should be investigated.
3. Fig1A – Why do non-specific bands (labelled with asterisks) in anti-YP3416 and anti-YP3418 blots disappear in Δy_p_{3416} and Δy_p_{3418} ?
4. Fig1A and 2F: The authors report lack of yp3416 secretion in Δy_p_{3418} cells. However, they can complement the knockout strain with a plasmid encoding yp3416. How can that be explained?
5. If the authors wish to rename yp3416 and yp3418, they should do so as early as possible in their manuscript to avoid having different gene / protein names in Figs1-3 and Figs4-7, respectively.
6. The text requires some clarification. Examples include the title, lines 132/133, line 262 (a mouse experiment is described here, not the bubonic plague)

Reviewer #3 (Remarks to the Author):

The manuscript entitled “Yersinia pestis E3 ligases ubiquitinate and degrade GBPs to subvert host defense - an arm acquired during evolution” reports an analysis of the role played by two genes coding for E3 ligases in Yersinia pestis’ weaponry against mammalian host defenses. The text is well written and pleasant to follow. The data are sound and convincing. Authors used multiple technologies to reach their goal: mutagenesis, immune-analyses, microscopy, generation of bacteria and mouse mutants etc. Together with the biochemical and cellular analysis of the processes controlled by these genes, the authors show that their functions differ in Y. pestis and Y. pseudotuberculosis and suggest that they diverged when Y. pestis acquired its high pathogenicity. The importance of inflammasomes inhibition in plague pathogenesis is a recently revealed mechanism and the present work brings interesting new elements. Although the work did not involve a Y. pestis pathogenic for humans (only Y. pestis microtus) and was limited to bubonic plague, it represents an important step in the understanding of events having led the plague bacillus to acquire more and more virulence.

Various points must be considered before publication:

- The Y. pestis biovar Microtus (used to document the role played by the genes in bubonic plague) is not pathogenic for humans. To generalize the conclusions to human plague, observations on virulence of mutants should be confirmed with a fully virulent strain (the widely used CO92 for example).
- The inhibition of inflammasome pathways is convincing, however the mutants show a limited attenuation of virulence in mice (mutants kill all of them, it’s only delayed), suggesting that inflammasome could play a limited role in host defense. Also, authors show that the E3 ligase - mediated inflammasome inhibition work in human cells, whereas humans resist to infection by Y. pestis microtus. This is surprising in the light of recent works on inflammasome inhibition by Y. pestis and the role of the inflammasomes pathways in host (mouse and humans) protection should be more discussed.
- Please clarify what are the ‘catalytic dead’ mutants: their defect is not explained, they first appear line 92 but their names are given only later, and their functional defect is not mentioned in Table S2. In that table and S3 as well, many strains listed do not appear in the text, so the list is in part useless. The reviewer suggests adding a column to the table, indicating the use of the strain.
- The absence of NEL domains or complete genes in Y. pseudotuberculosis is poorly discussed. The diversity of loci in this ancestor from which Y. pestis emerged suggests a loss by Y. pseudotuberculosis rather than a gene acquisition by Y. pestis. Please propose hypotheses.
- Title: ‘an arm’ or ‘arms’
- All figures: how many times were experiments repeated?

- In figure 2, what is strain '122'? It is not mentioned in the text nor in tables S2 & S3 listing the strains. Is it a mutant?
- In Figure 3D, the *Microtus* branch of the tree (represented by strain 91001) should be pointed since this biovar was used all over the work.

Response to referees

Reviewer #1

Previous studies have shown that guanylate-binding proteins (GBPs) have been shown to recognize pathogen-containing vacuoles, including the Yersinia-containing vacuole (YCV), leading to both lysis of this intracellular niche and induction of the inflammasome. However, the contributions of GBPs during in vivo infection with *Y. pestis*, the bacterium that causes the human disease known as the plague, has not been previously described. Here, Cao et al. present data suggesting that two newly identified genes in *Y. pestis* 201 (YP_3416 and YP_3418) help *Y. pestis* to evade recognition by GBPs and the innate immune system to promote host colonization. Both YP_3416 and YP_3418 have N-terminal LRR domains (a protein interaction domain) and C-terminal NEL domains (a Novel E3-ubiquitin Ligase domain). Using a combination of biochemical, cell culture, and animal infections, the authors present data supporting that: 1) YP_3416 and YP3418 are secreted into host cells via the Ysc type 3 secretion system, 2) infection of host cells by *Y. pestis* 201 results in ubiquitination and degradation of host guanylate-binding proteins (GBPs), 3) ubiquitination and degradation of host GBPs is dependent on YP_3416 and YP_3418, 4) GBPs inhibit intracellular survival of *Y. pestis* 201, 5) YP_3416 and YP_3418 inhibit GBPs recognition of the Yersinia-containing vacuole, 6) YP_3416 and YP_3418 contribute to virulence during bubonic plague, which is dispensable in the absence of mouse GBPs. The manuscript is clearly written, and provide several independent lines of evidence supporting the role of these proteins in *Y. pestis* virulence. I think these data represent several exciting discoveries that merit consideration for publication. First, YP_3416 and YP_3418 are chromosomally encoded, and to my knowledge represent the first effectors secreted by the Ycs T3SS that are not encoded on the pCD1 plasmid (which encodes the T3SS and the seven bonafide T3SS-dependent effectors secreted by *Y. pestis* known as the Yops). Second, these two genes are conserved in the highly virulent *Y. pestis* strains, but appear to be less conserved in *Y. pseudotuberculosis*, suggesting

that acquisition of YP_3416 and YP_3418 may have contributed to the transition of *Y. pestis* from a gastrointestinal pathogen to a blood-borne pathogen. Finally, the authors also present the first data indicating that GBPs contribute to resistance to *Y. pestis* infection, supporting the hypothesis that a transitional intracellular niche for *Y. pestis* during bubonic plague contributes to overall virulence of this facultative intracellular bacterium. That being said, the authors' conclusions would be greatly strengthened by the addition of replicate data for many of the figures and better description of statistics during animal studies.

Response: Thanks for your positive comments and suggestions that are very valuable and helpful for improving our manuscript. We have made careful corrections, especially provide replicate data for figures and statistical analysis results. We are glad to see that our manuscript has been greatly improved after these corrections, and we hope the revised manuscript will meet your requirement.

Figure 1: I appreciate using both in vitro and cell culture models to monitor T3SS-dependent translocation of YP_3416 and YP_3418. However, I am confused why some bands are highlighted as non-specific and marked with "*" when they are absent in the individual gene deletions. This would indicate to me they are not non-specific. Also, at 84% protein identity, how do you not have cross reactivity between 3416 and 3418? The methods do not describe if these are against full length proteins or only peptides that are specific for each protein.

Response: We sincerely appreciate your valuable comments and have conducted some experiment to determine the cross-reactivity between the two proteins.

1) The antibodies against YP_3416/YP_3418 were raised in rabbit using full length recombinant YP_3416/YP_3418 proteins. Equal amounts of recombinant YP_3416 and YP_3418 proteins were run on a 12% SDS-PAGE and transferred onto membrane, and proteins on the membrane were visualized using anti-YP_3416 or anti-YP_3418 antibodies followed by IRDye 800CW-conjugated goat anti-rabbit antibody to determine the cross-reactivity between the two antibodies.

The results showed that there was a relatively low level of cross-reactivity because the band showing the binding of anti-YP_3416 antibody to YP_3418 was much weaker than that of anti-YP_3418 and vice versa; thus, we think there should be limited influence on detection of YP_3416/YP_3418 in bacteria. Description on the immunogens used in preparation of anti-YP_3416 and anti-YP_3418 antibodies have been added in “Antibodies and reagents” section in the revised manuscript. Please refer to line 392-393.

- 2) We are sorry for the confusion caused by the original description and agree that the bands marked with “*” might come from some unknown protein complexes involved in YP_3416 or YP_3418. The statements of “non-specific” bands have been corrected. Please refer to the legend of Fig. 1.

Figure 2: All of these data in this figure is from one representative experiment. The rigor of these data would be significantly improved if you included quantification of bands relative to actin for multiple experiments, and include the statistics to support your observations.

Response: Thanks for your valuable suggestions. All the experiments shown in this figure have been conducted at least three times to get the stable and reproducible results. According to your suggestions, quantitative analysis was obtained using ImageJ software, the difference in protein levels were statistically analyzed and plotted using GraphPad Prism 8.0 software. The results were supplemented in Fig. 2 as panel H, I and J in the revised manuscript. Please refer to the legend of Fig. 2.

Figure 3 and Lines 130-140: In this section, you argue that *Y. pseudotuberculosis* YPIII is not able

degrade GBPs despite of the fact that it encodes homologs to YP_3416 and YP_3418. However, in Line 137 you state that this strain of YpIII lacks the pYV plasmid that encodes the T3SS. Why are you using a T3SS deficient strain for this comparison? Wouldn't a pYV positive strain be much more appropriate, and the only way to specifically show that the homolog is *Y. pseudotuberculosis* are not active. You should also need data from replicates for 3A and 3B.

Response: Thanks for your suggestions and we highly agree that pYV positive strain would be much more appropriate to make the comparison. In experiments shown in Fig. 3, we use YpIII strain for the following reasons. 1) A considerable fraction of *Y. pseudotuberculosis* strains harbor no T3SS-encoding pYV1 plasmid (12/23 strains in NCBI database), and the results using YpIII strains in infection experiments confirmed again that YP_3416 and YP_3418 of T3SS-negative strains cannot function in host cells. 2) A pYV positive strain pa3606 has been shown to be unable to degrade hGBPs in Fig. 2 already, and more importantly, we have no suitable pYV positive *Y. pseudotuberculosis* strains in hand to test their E3 ligase activity. To compensate for this deficiency, we performed the following experiment.

The *yp_3416* homolog in *Y. pseudotuberculosis* IP2666pIB1 is one of the proteins that are evolutionally closest to *yp_3416* amongst all the analyzed *Y. pseudotuberculosis* strains. To evaluate the capability of IP2666pIB1 in hGBPs degradation, we cloned the *Yp_3416~3418* homolog in IP2666pIB1 (*Yp_3416* and *Yp_3418* homologs were tagged with FLAG and Myc tag, respectively) into pACYC184 vector and the recombinant plasmid was transformed into $\Delta Yp_3416\sim 18$ (a 201 mutant with deletion of *yp_3416 ~yp_3418*, Cao S.Y., et. al, MCP, 2021) to get 201-E3YPT_{IP2666pIB1} strain. The expression of the E3 homologs in 201-E3YPT_{IP2666pIB1} was confirmed by immunoblotting using antibodies against FLAG- or Myc-tag. HeLa cells were infected with the wild-type *Y. pestis* strain 201 or 201-E3YPT_{IP2666pIB1} and we found that 201-E3YPT_{IP2666pIB1} infection cannot degrade hGBPs, in sharp contrast to the significant degradation of hGBPs by 201 infection. Since *Y. pestis* expressing E3 homolog in IP2666pIB1, which is closest to *Y. pestis*, cannot degrade GBPs, we

reckoned that all the *Y. pseudotuberculosis* strains cannot degrade hGBPs during infection. These results have been supplemented in the revised manuscript as Fig. 3F and I. Please refer to line 164 to 176.

Quantitative analysis was obtained using ImageJ software, the difference in protein levels were statistically analyzed and plotted using GraphPad Prism 8.0 software for Fig. 3 A, C and F and shown as Fig. 3 G, H and I, respectively, in the revised manuscript.

Figure 4: All of these western blot data in this figure is from one representative experiment. The rigor of these data would be significantly improved if you included quantification of bands relative to actin for multiple experiments, and include the statistics to support your observations.

Response: Thanks for your suggestions. All the experiments shown in this figure have been conducted at least three times to get the stable and reproducible results. We have used Image J to obtain the quantitative results from three independent experiments, the difference in protein levels were statistically analyzed and plotted using GraphPad Prism 8.0 software for Figure 4 A-F, P and Q, and shown as Figure 4 R, S and T, respectively, in the revised manuscript.

Figure 5: I am a little concerned by the interpretation of this figure. I am not sure you can conclude that GBP1 co-localizes with *Y. pestis*. To conclude this, I think you need to generate a Pearson correlation coefficient in order to have some statistical analysis to back this up. It appears that hGBB1 is everywhere in the cell and the reason there is not a colocalized signal is because it is either not as well expressed or degraded during *Y. pestis* infection (latter is more likely based on your other data). Moreover, bacteria were grown under conditions that should induce the T3SS and inhibit phagocytosis. This would suggest that the majority of bacteria should be extracellular. Does this mean that the GBPs are being recruited to the plasma membrane under attached bacteria? If YPIII is lacking pYV as described in Line 137, then is there significantly more intracellular bacteria than in the *Y. pestis* 201

infected situation? Again, does the lack of the T3SS artificially result in a lack of degradation of GBP?

Response: Thanks for your valuable comments.

- 1) All the bacterial strains in this experiment were cultured at 26 °C in LB medium, under which the expression of anti-phagocytotic mechanisms including T3SS and F1 capsule are relatively low. Thus, *Yersinia* bacteria are prone to be taken up by the cells after being added into the cultures, and most of them should be inside the cells rather than attaching to the cell surfaces. Actually, we usually have to add Cytochalasin D to HeLa cell cultures to inhibit the internalization of *Y. pestis* bacteria (Tan, Y.F. et. al, 2017, IAI; Wang, T. et. al, 2018, AEM).
- 2) We have calculated Pearson correlation coefficient for these images using ImageJ and statistical analysis has been included in the revised manuscript. HeLa cells stably expressing GFP-hGBP1 were used in this experiment and expression level of hGBP1 were the same in different samples. The total GFP-hGBP1 signal intensity was very low in cells infected with the wild type *Y. pestis* in Fig. 5 at the time of taking these images (2.5 hpi.), because hGBP1 was almost completely degraded by YspE1/YspE2, and similar results can be found in Fig. 3. Therefore, only a few co-localization signals of the wild type *Y. pestis* 201 with GBP1 could be observed. For the dynamic changing process, please refer to the Video S1 and S2.
- 3) With regard to *Y. pseudotuberculosis* strain YpIII in Fig. 5, we agree that lacking of pYV1 plasmid will directly lead to the failure of GBP degradation, irrelevant to the function of homolog of *yp_3416~3418* locus in YpIII, but due to the defection in T3SS. However, as that has been mentioned in answer to the question about Fig. 3, we have a limited number of pYV positive *Y. pseudotuberculosis* strains to perform this comparative experiment. To compensate this deficiency, we examined whether 201-E3YPT_{IP2666pIB1}, a *Y. pestis* strain expressing E3 homologue in IP2666pIB1 that is closest to *Y. pestis*, can degrade GBPs. The results showed that GBPs were quite stable in cells infected with 201-E3YPT_{IP2666pIB1} (Fig. 3F and I). Therefore, YpIII is only a suboptimal selection for us when a suitable *Y. pseudotuberculosis* strain is unavailable.

Figure 6: If the degradation of GBPs is the only function of YP_3416 and YP_3418 then wouldn't you expect that the phenotypes would overlap in D and E? This was not clearly addressed in the manuscript. The rigor of the data in G, H, and I would be significantly improved if you included quantification of bands relative to actin for multiple experiments, and include the statistics to support your observations. The observed decrease in intracellular survival as compared to previous papers that the authors discussed in Lines 297-305 is likely due to the extremely high dose of, and extended incubation period with, gentamicin that was employed in these assays (see PMID: 29312891). Repeating these assays with a lower amount of antibiotic may further differentiate a GBP/ YP_3418 phenotype in the macrophages.

Response: Thanks for your valuable comments. We highly agree that the different concentrations of gentamicin will indeed influence the results of intracellular survival. As suggested by the reviewer, we repeated the intracellular survival assays using a lower amount of gentamicin (30 µg/ml instead of 100 µg/ml for killing of extracellular bacteria and 2 µg/ml for inhibition of growth after 1 hpi.) for three independent times. Different from the experiment results that no enrichment of bacterial numbers was observed when high dose of gentamicin was used, a significant replication has been found for both 201 and Δyp_3418 at 2 hpi., and the extent was significantly less for Δyp_3418 (Fig. 6C). Although the two survival curves have not overlapped completely, no significant difference in survival in $Gbp^{chr3^{-/-},chr5^{-/-}}$ BMDMs was found between 201 and Δyp_3418 . (Fig. 6D and F). However, in IFN- γ -treated $Gbp^{chr3^{-/-},chr5^{-/-}}$ BMDMs, a significant difference in survival between 201 and Δyp_3418 was observed at 4 hpi., suggesting that some IFN- γ induced cellular targets other than GBPs might be present and plays roles in *Y. pestis* pathogenesis. Please refer to line 246 to 259 in the revised manuscript.

Quantitative analysis was obtained through the ImageJ software, the difference in protein levels were statistically analyzed and plotted using GraphPad Prism 8.0 software for Fig. 6 H, I and J and shown as Fig. 6 K and L, respectively, in the revised manuscript.

Figure 7: Survival curves lack statistical analysis. Are these curves significantly different? The most likely tissue that the bacteria will be intracellular are the lymph nodes. Also, the lymph nodes are a bottle neck for subsequent dissemination, but colonization of these tissues are missing. Therefore, a kinetic time course analysis of lymph node colonization in WT and KO mice would greatly increase the authors' hypothesis that YP_3418 contributes to intracellular survival that improves virulence. Legend is missing group size, sex of mice, and number of times the experiment was repeated (should be repeated to show reproducibility).

Response: Thanks for your insightful suggestions.

- 1) Statistical analysis for survival curves in Figure 7 has been done and the corresponding results were supplemented in the revised manuscript. Please refer to line 287-289, line 300-303 and the legend of Fig. 7 in the revised manuscript.
- 2) We highly agree that the lymph nodes are a bottle neck for subsequent dissemination. As suggested by the reviewer, we repeated experiments of bacterial colonization in tissues including lymph nodes for two independent times in WT mice and three times in *Gbp^{chr3-/-, 5-/-}* mice, respectively, and 10 female mice (6-8 weeks) per group for each of the bacterial strains were used. The two or three independent replicate experiments have acquired similar results and only one set of data was shown in Figure 7B and D. Significant difference was found in lymph nodes between 201- and ΔYp_{3418} -infected C57 mice. Colonization of ΔYp_{3418} in the lymph nodes of *Gbp^{chr3-/-, 5-/-}* mice enhanced significantly ($p < 0.05$) and an even higher enhancement occurred in liver ($p < 0.0001$), implying that an increased colonization in lymph nodes greatly prompted the subsequent dissemination of *Y. pestis* (Figure 7D). Please refer to line 289 to 290 and line 305 to 309.

- 3) The median survival day of *Gbp*^{chr3^{-/-}, 5^{-/-}} mice challenged with ~100 CFU of WT *Y. pestis* was 3.5 to 4, and a substantial fraction of mice died at 3 days post infection, thus we collected the tissues from mice at about 60~72 h post infection (2.5~3 days). In another work we have done, 10⁶ CFU of WT *Y. pestis* 201 cells were subcutaneously inoculated into mice and we cannot detect substantial numbers of lymph cells containing *Y. pestis* until 24 h (unpublished), and no living bacteria will be detected at this time point if the inoculation dose is lowered to 100 CFU as that was used in this study. Thus, it's hard to analyze the dynamic of living bacterial numbers in lymph node at the challenge dose of ~100 CFU in a relatively short period of time (LD₅₀ of 201 for mice is about 3 CFU, Yang, F.K., PLoS ONE, 2010).
- 4) Legend for Figure 7 has been supplemented with group size, sex of mice and the statistical analysis results in the revised manuscript.

Minor Comments:

Line 174: Where is binding data for 1, 2, 5, and 6?

Response: These data were shown in Fig S4 (A~D) and they have been cited in the revised manuscript. Please refer to line 191.

Line 191: Adding the KD for IpaH9.8 would be helpful to provide context.

Response: Thanks for your suggestion and the KD for IpaH9.8 has been added in the revised manuscript. Please refer to line 208.

Reviewer #2

The authors report the characterization of YspE1 and YspE2, two novel ubiquitin E3 ligases of the NEL class in *Yersinia*, which the authors demonstrate degrade several GBPs. The role of YspEs therefore appears similar to the related NEL ligase IpaH9.8 in *Shigella*, which is known to also antagonize GBP function through ubiquitylation (Li Nature 2017; Wandel, Cell Host Microb 2017). Similar to IpaH9.8, YspE2 also forms K48 linked chains on GBP1 that result in proteasomal degradation of GBPs. Deletion of YspE1s caused enhanced GBP association with *Yersinia*, improved anti-bacterial activity of host cells in vitro and reduced mortality in infected mice, again similar to the situation in *Shigella* (Wandel Nat Immunol 2020).

Taken together, while the report of two novel NEL class E3 ligases in *Yersinia* and the identification of GBPs as their substrate is interesting, particularly to microbiologist in the *Yersinia* field, in more general terms the manuscript merely confirms the important role of GBPs in anti-bacterial immunity and that bacteria overcome GBP-mediated defence through ubiquitin-dependent proteolysis by providing another example in a different pathogenic bacterium.

Response: Thanks for your precious comments and they are very valuable and helpful for us to improve our manuscript as well as to our future studies. We have made careful corrections according to your suggestions and hope the revised manuscript will meet your requirements.

Major points:

1. Line 210 – Using epithelial cells, the authors report that “...GBP1 was largely co-localized with” *Yersinia*. The authors should determine the fraction of GBP1-coated bacteria, as well as the fraction of cytosol exposed bacteria. Is the majority of *Yersinia* accessing the cytosol and are GBPs directly recruited to bacteria? Or are GBPs accumulating on *Yersinia* containing vacuoles? Higher resolution

images as well as quantification of marker colocalization will be needed. Galectin containing might be useful.

Response: Thanks for the reviewer's insightful comments. We have calculated Pearson correlation coefficient for results shown in Figure 5, and we estimated that 69~76% bacteria expressing YspE2 were colocalized with hGBP1 using the colocalization percentage of red fluorescence with green fluorescence signals. However, we are not sure the fraction of cytosol exposed bacteria. Inspired by the reviewer's advice, we performed infection experiment in HeLa cells stably expressing GFP-hGBP1, and analyzed the colocalization between hGBP1 and *Y. pestis* bacteria, as well as the Galectin-3 and Galectin-8 distributions post infection. Unfortunately, we cannot observe Galectin labeled vacuole till now possibly due to the technique or some other problems in our experimental system. Besides, epithelial cells like HeLa might have limited ability to form bacterial containing vacuole and thus is not an ideal system to do such experiment. Human or murine macrophages cells stably GFP-hGBP1 should be help, I think we will try to do these experiments in our future study. Please refer to the revised Figure 5.

2. The degradation of endogenous GBPs (or at least selected GBPs) upon Yersinia infection should be investigated.

Response: Thanks for the reviewer's valuable advice. We have determined the endogenous GBP1 in U937, HeLa and RAW264.7 cells post *Y. pestis* infection. Cells were primed with IFN- γ followed by infection with the wild type *Y. pestis* 201 or Δy_p_{3418} . After 2 hours' infection, the level of endogenous hGBP1 or mGBP1 was analyzed by immunoblotting and the results indicated that GBP1 can be degraded by strain 201 in all the three types of cells. These results have been supplemented in Fig. 3C, H and Fig. 4Q, T in the revised manuscript. Please refer to lines 148-150 and 216-217.

3. Fig1A – Why do non-specific bands (labelled with asterisks) in anti-YP3416 and anti-YP3418 blots disappear in $\Delta y p_{3416}$ and $\Delta y p_{3418}$?

Response: We are sorry for the confusion caused by the incorrect descriptions and we deem that the bands marked with “*” might come from some unknown protein complexes involved in YP_3416 or YP_3418 rather than non-specific bands. The statements of “non-specific” bands have been corrected. Please refer to the legend of Fig. 1.

4. Fig1A and 2F: The authors report lack of yp3416 secretion in $\Delta y p_{3418}$ cells. However, they can complement the knockout strain with a plasmid encoding yp3416. How can that be explained?

Response: Thanks for the reviewer’s insightful comments and this is an interesting question. The phenomenon that $\Delta y p_{3418}$ bacteria cannot secrete both YP_3416 and YP_3418 but $\Delta y p_{3416}$ can normally secrete YP_3418 suggested that YP_3418 might facilitate the delivery of YP_3416 via an yet unknown mechanism. Due to that YP_3418 and YP_3416 have more than 83.3% identities, we assume that YP_3416 could possibly be able to act as a functional substitute for YP_3418. Thus, we cloned *yp_3416* and *yp_3418* into pACYC184 and complemented $\Delta y p_{3418}$ strain with pACYC184-*yp_3416* and pACYC184-*yp_3418* to obtain $\Delta y p_{3418}/YspE1$ and $\Delta y p_{3418}/YspE2$, respectively. Cell infection experiment results showed that the two complemented strains have comparable GBP1 degradation capability, confirming our assumption that YP_3416 and YP_3418 showed similar function. Furthermore, results of experiment examining the substrates preference of YP_3418 and YP_3416 also showed that the two proteins have completely identical substrate specificities toward both the human and mouse GBPs.

As to how was YP_3416 delivered by $\Delta y p_{3418}/YspE1$ into the host cells in the absence of YP_3418, if the delivery of YP_3416 needs the aids of YP_3418, we cannot give an exact explanation. We guess that complementation of $\Delta y p_{3418}$ with pACYC184-*yp_3416* could result in a higher expression of

YP_3416 than the physiological level of this protein, add to that YP_3418 and YP_3416 have more than 83.3% identities, which probably promotes the translocation of YP_3416 into the host cytosol in a suboptimal condition for translocation (in the absence of YP_3418).

5. If the authors wish to rename yp3416 and yp3418, they should do so as early as possible in their manuscript to avoid having different gene / protein names in Figs1-3 and Figs4-7, respectively.

Response: Thanks for the reviewer's helpful suggestion. We have unified the protein names throughout the manuscript including Figs 1 to 3 and all the modifications have been highlighted in yellow in the revised manuscript.

6. The text requires some clarification. Examples include the title, lines 132/133, line 262 (a mouse experiment is described here, not the bubonic plague)

Response: Thanks for the reviewer's comments. As suggested by the reviewer, we have made corresponding clarifications. Line 134 has been modified to "*Y. pestis* infection leads to degradation of multiple hGBPs while *Y. pseudotuberculosis* is unable to dampen this defense mechanism". Line 282 has been modified to "YspE1/2 degradation of GBPs play a significant role in disease progress in a mouse model of bubonic plague".

Reviewer #3

The manuscript entitled "Yersinia pestis E3 ligases ubiquitinate and degrade GBPs to subvert host defense - an arm acquired during evolution" reports an analysis of the role played by two genes coding for E3 ligases in Yersinia pestis' weaponry against mammalian host defenses. The text is well written and pleasant to follow. The data are sound and convincing. Authors used multiple technologies to reach their goal: mutagenesis, immune-analyses, microscopy, generation of bacteria and mouse

mutants etc. Together with the biochemical and cellular analysis of the processes controlled by these genes, the authors show that their functions differ in *Y. pestis* and *Y. pseudotuberculosis* and suggest that they diverged when *Y. pestis* acquired its high pathogenicity. The importance of inflammasomes inhibition in plague pathogenesis is a recently revealed mechanism and the present work brings interesting new elements. Although the work did not involve a *Y. pestis* pathogenic for humans (only *Y. pestis* microtus) and was limited to bubonic plague, it represents an important step in the understanding of events having led the plague bacillus to acquire more and more virulence.

Response: Thank you for your positive comments and pointing out the significance of our work for the understanding of the virulence of plague pathogen. We highly appreciate your valuable suggestions, according to which we have made a lot of corrections, and we hope the revised manuscript will meet your requirements.

Various points must be considered before publication:

- The *Y. pestis* biovar Microtus (used to document the role played by the genes in bubonic plague) is not pathogenic for humans. To generalize the conclusions to human plague, observations on virulence of mutants should be confirmed with a fully virulent strain (the widely used CO92 for example).

Response: Thank the reviewer's comments and we totally understand your concern. Rodents are the major natural reservoirs of *Y. pestis* and human beings are only accidental hosts, tiny virulence changes of *Y. pestis* strains in rodent hosts will be under the natural selection pressure during evolution and features benefiting the survival or transmission of *Y. pestis* would be kept and stabilized thereafter. This study focused on the finding and confirmation of novel virulence mechanisms of GBPs degradation by *Y. pestis*, and revealed that this mechanism are newly acquired during evolution. We think that without functional confirmation of YspE1/2 in a fully virulent strain should not affect our main findings; and more importantly, the use of fully virulent human pathogenic *Y. pestis* strains in scientific experiments is under strict control in China due to biosafety concerns, and it's actually

impossible for us to do so. The 201 strain we used in this study is as virulent for mice ($LD_{50}=3$ CFU, *s.c.* infection of BALB/c mice) as human pathogenic *Y. pestis* strains such as CO92.

- The inhibition of inflammasome pathways is convincing, however the mutants show a limited attenuation of virulence in mice (mutants kill all of them, it's only delayed), suggesting that inflammasome could play a limited role in host defense. Also, authors show that the E3 ligase - mediated inflammasome inhibition work in human cells, whereas humans resist to infection by *Y. pestis* microtus. This is surprising in the light of recent works on inflammasome inhibition by *Y. pestis* and the role of the inflammasomes pathways in host (mouse and humans) protection should be more discussed.

Response: Thank the reviewer's positive comments and we will answer your questions in the following aspects.

- 1) We have not determined the LD_{50} of Δyp_3418 in mice to quantitatively assess its virulence attenuation. The challenge doses of *Y. pestis* for survival curves are about 100 CFU (the actual doses were determined by plating the serial diluted solutions of bacteria suspensions used in animal infection experiment and counting the number of living bacteria) for both of the wild type ($LD_{50}=3$ for BALB/c mice) and the *yp_3418* mutant. All the mice were killed by the *yp_3418* mutant although the medial survival days were significant longer, suggesting that LD_{50} for Δyp_3418 is less than 100 CFU but should be larger than 3 CFU by several to dozens of folds.
- 2) In cell infection experiments using both human cell lines U937 and HeLa cells, as well as the murine RAW264.7 cells and BMDMs, we could hardly detect the cleavage of caspase-1 and IL-1 β /IL-18 secretion post infection with the wild type 201 strain. No significant difference of inflammasome response to microtus strain 201 has been found between the human and murine cells, at least in cell infection experiments. We speculate that the host specific pathogenicity of

microtus strain 201 (highly virulent to mice but avirulent to human) might involve biological process other than inflammasome pathway.

3) We deem that inflammasome might not play roles in host defense against plague as important as we think. As mentioned in the reviewer's comment, although Δyp_3418 cannot inhibit inflammasome activation as the wild type 201 strain does, it shows a limited attenuation of virulence in mice. The sensors of inflammasome pathways (including NLRP3, NLRC4, Pyrin and so on) mainly reside in cytosol to detect PAPMs or DAMPs. *Y. pestis* is a facultative intracellular pathogen that initially replicates in host macrophages, but soon becomes phagocytosis-resistance and lives an extracellular life thereafter. Thus, compared to the obligate intracellular bacterial pathogens, inflammasome might not be so important for the host defense against *Y. pestis* infection. In a recent study showing that familial Mediterranean fever (FMF) mutations in human pyrin confer heightened resistance to *Y. pestis* (Yong H.P. et. al., NI, 2020), MefvM^{680I/M680I} FMF knock-in mice exhibited IL-1 β -dependent increased survival relative to wild-type knock-in mice, but were not totally resistant to *Y. pestis* infection (100 to 150 CFU per mouse, *Y. pestis* KIM5 that lack *pgm* locus was used), implying host defense mechanism other than inflammasome are playing critical roles in restriction of *Y. pestis* infection.

- Please clarify what are the 'catalytic dead' mutants: their defect is not explained, they first appear line 92 but their names are given only later, and their functional defect is not mentioned in Table S2. In that table and S3 as well, many strains listed do not appear in the text, so the list is in part useless. The reviewer suggests adding a column to the table, indicating the use of the strain.

Response: Thanks for the reviewer's kind reminding. We are sorry for the unclear description and the corresponding explanations have been supplemented in the revised manuscript.

- 1) The ‘catalytic dead’ mutants lost the E3 ligase activity due to the disruption of the critical residues C407 in YspE1 and C386 in YspE2 NEL domains, respectively. We have added these descriptions to clarify the defect of those mutants. Please refer to line 93 to 96.
- 2) Table S2 and S3 listed the *Y. pestis* and *Y. pseudotuberculosis* strains (complete genomes are available in the NCBI database) that have been subjected to multiple sequences alignment analysis in this study. They have not been used in any experiments of this study; therefore, they were not mentioned individually, but cited as a whole in the main text (Please refer to line 155 and 160).

- The absence of NEL domains or complete genes in *Y. pseudotuberculosis* is poorly discussed. The diversity of loci in this ancestor from which *Y. pestis* emerged suggests a loss by *Y. pseudotuberculosis* rather than a gene acquisition by *Y. pestis*. Please propose hypotheses.

Response: Thanks for the reviewer’s insightful comments. Homologs of *yp_3416~yp_3418* in *Y. pseudotuberculosis* strains often contains only one or two genes and some homologues lack NEL domain, thus they cannot utilize E3 ligase activity to disrupt the host defense mediated by GBPs. The diversity of *yp_3416~3418* loci suggested it’s not important or essential for this enteropathogen. In contrast, all the modern *Y. pestis* strains contains intact *yp_3416~yp_3418* loci and encode active YspE1/YspE2 E3 ligases. We hypothesize that complete *yp_3416~yp_3418* loci might be gradually formed (probably by gene duplication, only our conjecture) during evolution from *Y. pseudotuberculosis*, and harboring functional YspE1/YspE2 E3 ligases yields survival benefit or fitness in host to adapt to the new flea-borne transmission of the newly formed specie *Y. pestis*, whereby they were subsequently stabilized due to competitive advantages. We have added these discussions in the revised manuscript. Please refer to line 357 to 368.

- Title: ‘an arm’ or ‘arms’

Response: Thank the reviewer's suggestion. The title has been revised to “.....—— an arm acquired during evolution”.

- All figures: how many times were experiments repeated?

Response: At least three independent experiments have been performed for Figs 1 to 7 to get the stable and reproducible results, except results shown in Fig. 7B, in which bacterial colonization in tissues including lymph nodes according to a reviewer's comments were performed for two independent times in WT mice. For the immunoblotting assays for Fig. 2 to Fig.6, quantitative results from multiple experiments and statistical analysis have been added in the revised manuscript.

- In figure 2, what is strain '122'? It is not mentioned in the text nor in tables S2 & S3 listing the strains. Is it a mutant?

Response: We are sorry for this careless mistake and “122” in Fig. 2 has been corrected to “201”, the wild type *Y. pestis* strain used in this study. Please refer to line 376 to 377 and Table S1 for the descriptions of this strain.

- In Figure 3D, the *Microtus* branch of the tree (represented by strain 91001) should be pointed since this biovar was used all over the work.

Response: Thanks for the reviewer's helpful advice. We have labeled the representative strain 91001 in the phylogenetic tree.

References Cited

1. Shiyang Cao, Yuling Chen, Yanfeng Yan, Songbiao Zhu, Yafang Tan, Tong Wang, Yajun Song, Haiteng Deng, Ruifu Yang, and Zongmin Du. Secretome and Comparative Proteomics of *Yersinia*

pestis Identify Two Novel E3 Ubiquitin Ligases That Contribute to Plague Virulence. *Molecular Cellular Proteomics*, 2021, 20, 100066.

2. Tong Wang, Min Wang, Qingwen Zhang, Shiyang Cao, Xiang Li, Zhizhen Qi, Yafang Tan, Yang You, Yujing Bi, Yajun Song, Ruifu Yang, Zongmin Du. Reversible Gene Expression Control in *Yersinia pestis* by Using an Optimized CRISPR Interference System. *Applied and Environmental Microbiology*, 2019, 85(12): e00097-19.
3. Yafang Tan, Wangbing Liu, Qingwen Zhang, Shiyang Cao, Haihong Zhao, Tong Wang, Zhizhen Qi, Yanping Han, Yajun Song, Xiaoyi Wang, Ruifu Yang, Zongmin Du. 2017. *Yersinia pestis* YopK inhibits bacterial adhesion to host cells by binding to the extracellular matrix adaptor protein matrilin-2. *Infection and Immunity*. 2017, 85(8). doi:10.1128/IAI.01069-16.
4. Fengkun Yang, Yafang Tan, Yuehua Ke, Yujing Bi, Qinghai Shi, Huiying Yang, Jinfu Qiu, Xiaoyi Wang, Zhaobiao Guo, Hong Ling, Ruifu Yang and Zongmin Du. Cell membrane is impaired, accompanied by enhanced type III secretion system expression in *Yersinia pestis* deficient in RovA regulator. *PLoS One*. 2010 (5): e12840.

Reviewers' comments:

Reviewer #1 (Remarks to the Author):

The authors added important data and additional clarification to the revised manuscript. However, I think there are still some issues with revised manuscript that need to be resolved.

1. Thank you for including the replicate number and data in all of your figures. Inclusion of these data alleviate any concerns about rigor.

2. Lines 134-135; Fig. 3: I appreciate the additional description of the homology of these proteins in *Y. pestis* and *Y. pstb*. However, I think there are still some problems with this section that do not fully support the conclusions made in this section. See below:

a. I strongly recommend that you remove the YPIII data. The YPIII strain you are using is missing pYV and thus the T3SS. Without the T3SS, this strain does not provide any direct data about the activity of the YspE1 and YspE2 homologs in YPIII (the homologs cannot be translocated into the host cells by this strain). Even stating that the phenotype is dependent on the T3SS is misleading because you later argue that *Y. pstb* homologs are not active. Its inclusion will likely just confuse the reader, weakening your overall arguments.

b. I appreciate the complementation strategy with the IP2666pIB1 YspE1 homolog. However, the *Y. pestis* strain used for the complementation strategy lacks yspE2 and you previously demonstrated in Figure 1 that a yspE2 mutant does not translocate YspE1. Therefore, one could conclude that the IP2666pIB1 YspE1 homolog is also not translocated by the yp_3416-yp_3418 mutant and your data are due to an artifact of this translocation data. Without demonstrating translocation, these data become suspect. Therefore, translocation data needs to be included. Alternatively, transfection of eukaryotic cells with the IP2666pIB1 YspE1 homolog would be a more direct test of whether this homolog can degrade GBPs (as you did in Figure 4)?

c. What is the identity between YspE1 and the IP2666pIB1 homology? It appears to be >85%, which is significantly higher than the identity to the homologs in *Shigella* that were used to first generate your hypothesis that these proteins even target GBPs. Is divergence occurring in the LRR domain that might predict different targets or in the NEL catalytic domain, which might indicate it is inactive? A discussion of this should be included somewhere.

3. Lines 220-237; Fig. 5: I strongly recommend removing the YPIII data from this section for the same reasons as stated above.

4. Lines 220-237, Fig. 5: I am still concerned with the statement that you are analyzing intracellular bacteria in Fig. 5. Based on the description in the methods (lines 505-517), one would still predict the majority of the *Y. pestis* should be extracellular in this assay – methods state that bacteria were grown at 26 then shifted to 37 for 2 h prior to infection (this would induce the T3SS and inhibit phagocytosis, especially during infection of HeLa cells that are not professional phagocytes). The exception might be YPIII, which might still be expressing Inv to promote uptake. What percentage of these bacteria are taken up by HeLa cells (these are not professional phagocytes) and what percentage are simply adhered to the cells? I assume the figures are compressed Z stacks (though not mentioned in the legend). If these were intracellular bacteria in a phagosome, slices should show a ring of GBP around the bacteria. As previously mentioned by another reviewer, higher magnification/resolution images would help demonstrate this (images should include scale bars). Also, doing an ANOVA on Pearson's Correlation Coefficient is not appropriate. The Pearson's Correlation Coefficient measures the strength and direction of a linear relationship between two variables. For colocalization studies, the values range between 0 and 1, with 1 indicating a direct correlation (more likely to colocalize) and 0 indicating no correlation (more likely to not colocalize). By adding the ANOVA, you are doing a statistical analysis on a statistical analysis. Reporting the Pearson's Correlation Coefficients is sufficient.

5. Line 295: "Hypersusceptible" seems to be an over statement of the data. There is only a 24 h difference when all animals reach end point and the mean time death appears to be equal for both

strains of mice infected with 201. There is likely no difference in LD50 between the mouse backgrounds. It seems that “more susceptible” is a better description of the phenotype. Moreover, because this phenotype is so subtle, it seems that the data from the second independent experiment should be included/reported to ensure that this is a reproducible phenotype. I agree that there might be differences in susceptibility between the 3418 mutant, but this data does not convince me that these mice are hypersusceptible to WT *Y. pestis* infection.

6. Line 351-353: For reasons stated above, I do not believe the authors can conclude that they have demonstrated that the proteins from Pa3606 can degrade the GBPs. The case of YPIII is even weaker, as all of the data from this strain was performed with a strain that lacks the ability to secrete the proteins into host cells.

7. Line 369-370: The data do not definitively show that the *Y pstb* homologs do not degrade the GBPs.

Additional Minor comments:

1. The switching of between *yp-3416/3418* and *ypE1/E2* has become more confusing in the new draft (even individual figures use both designations). Suggest that you standardize nomenclature throughout the manuscript

2. Lines 274-281, Fig. 6J: I had to reread this section a couple times before it was clear that you ultimately concluding that you could not detect differences in inflammasome activation in macrophages from chr 3/5 mice. During my first read, it appeared that you were over interpreting your data in Fig 6J with the conclusion that inflammasome activation is even occurring in these cells. Based on the representative image for 6J, Casp 1 expression is extremely low in macrophages from chr 3/5 mice (see mock treated cells), and it appears that a much longer exposure was required to see any bands in infected cells (based on the greater appearance of secondary bands in cell lysate and supernatant). Perhaps it would be better to simplify this statement. For example, “Due to low levels of Casp 1 in macrophages from Chr 3/5 mice, we were unable to detect significant inflammasome activation in these cells during *Y. pestis* infection (Fig. 6J). I don’t think these cells are required to make the overall conclusion of this section.

3. Fig 6: This is the first figure that you designate *Y. pestis* 201 as WT. All previous figures use the 201 designation. Recommend that you use consistent nomenclature throughout to reduce confusion by readers.

4. Fig. 6: How long were macrophages incubated with 30 ug/ml gent. before changing to 2 ug/ml and how does this relate to the time points in figure 6 (does 2 h post infection reflect the time at which the 30 ug was removed)? This should be clear for others who might want to reproduce your data, but is also important for the reader to figure out total bacteria added vs. associated vs. intracellular.

5. Lines 289-291; Fig. 7: Include a description of when these tissues were harvested and bacterial burdens were determined. Fig. 7B and D require inclusion of the limit of detection for each tissue. Authors may also want to consider changing this to a box and whisker graph with all points, as it might better represent the differences.

6. Fig. 7B. It seems that a one way ANOVA is the more appropriate analysis (there is only one variable, the strain used to infect the animals).

7. Fig. 7D: This data would be easier for the reader to interpret if it was grouped by tissue not mouse strain (similar to 7B).

8. Line 263-266: It is not stated anywhere how long the infections took place before they were measured for inflammasome activation. Would later time points show activation?

Reviewer #2 (Remarks to the Author):

The authors have made reasonable efforts to address my questions.

However, questions remain:

1.) The ability of NEL ligases (specifically IpaH9.8) to degrade GBPs and to antagonize host immunity is well established (Li Nature 2017, Wandel Cell Host Microbe 2017, Wandel Nat Immunol 2020). Despite pointing out these points in my original summary of their paper, the authors keep avoiding the issue. In my opinion, these points need to be addressed straightforwardly in the introduction and should not be hidden somewhere in the text.

2.) Former Major Point 1: It is regrettable that the authors failed to quantify the extent to which Yersinia invades the host cytosol. Galectins are well established tools and work without problems in HeLa cells. Have they tried any positive controls (Listeria, Shigella, Salmonella) to test their galectin constructs and did they try to stain endogenous galectins if their expression system does not work? Regarding the colocalization of GBPs and Yersinia – is the Pearson's coefficient now reported in Fig5F really informing us about the percentage of bacteria associated with GBPs as claimed by the authors in their rebuttal?

3.) The title is still non-sensical.

Reviewer #3 (Remarks to the Author):

Authors have made a clear effort to answer all the points raised and the manuscript is now acceptable for publication.

Response to Referees

Reviewers' comments:

Reviewer #1 (Remarks to the Author):

The authors added important data and additional clarification to the revised manuscript. However, I think there are still some issues with revised manuscript that need to be resolved.

1. Thank you for including the replicate number and data in all of your figures. Inclusion of these data alleviate any concerns about rigor.

Response: We sincerely appreciate your positive comments on the rigor of our data and all the valuable comments you have made on our manuscript.

2. Lines 134-135; Fig. 3: I appreciate the additional description of the homology of these proteins in *Y. pestis* and *Y. pstb*. However, I think there are still some problems with this section that do not fully support the conclusions made in this section. See below:

a. I strongly recommend that you remove the YPIII data. The YPIII strain you are using is missing pYV and thus the T3SS. Without the T3SS, this strain does not provide any direct data about the activity of the YspE1 and YspE2 homologs in YPIII (the homologs cannot be translocated into the host cells by this strain). Even stating that the phenotype is dependent on the T3SS is misleading because you later argue that *Y. pstb* homologs are not active. Its inclusion will likely just confuse the reader, weakening your overall arguments.

Response: Thanks for your helpful advices. We admit that the YPIII data is misleading and has not been well presented and interpreted. It has been removed according to your suggestion.

b. I appreciate the complementation strategy with the IP2666pIB1 YspE1 homolog. However, the *Y. pestis* strain used for the complementation strategy lacks yspE2 and you previously demonstrated in Figure 1 that a yspE2 mutant does not translocate YspE1. Therefore, one could conclude that the IP2666pIB1 YspE1 homolog is also not translocated by the yp_3416-yp_3418 mutant and your data are due to an artifact of this translocation data. Without demonstrating translocation, these data become

suspect. Therefore, translocation data needs to be included. Alternatively, transfection of eukaryotic cells with the IP2666pIB1 YspE1 homolog would be a more direct test of whether this homolog can degrade GBPs (as you did in Figure 4)?

Response: Thanks for your appreciation of our complementation strategy with the IP2666pIB1 YspE1 homolog. We agree that the translocation of IP2666pIB1 YspE1 homolog must be demonstrated before we can conclude that it is unable to degrade hGBP1. According to your suggestion, we removed these results and using transfection assays to determine whether these homologs can degrade hGBPs. Plasmids expressing the YspE1/YspE2 homolog of IP2666pIB1 were transfected into HeLa cells stably expressing hGBP1 and the level of hGBP1 was measured. The results showed that hGBP1 can be degraded in cells expressing YspE1 but not YspE1/YspE2 homolog of IP2666pIB1. Please refer to lines 158-165. These results have been presented in Figure 4 in the revised manuscript.

c. What is the identity between YspE1 and the IP2666pIB1 homology? It appears to be >85%, which is significantly higher than the identity to the homologs in *Shigella* that were used to first generate your hypothesis that these proteins even target GBPs. Is divergence occurring in the LRR domain that might predict different targets or in the NEL catalytic domain, which might indicate it is inactive? A discussion of this should be included somewhere.

Response: We are grateful for your insightful comments. Sequence identity between YspE1 and IP2666pIB1 homolog is indeed over 85% as revealed by sequence alignment, a much higher identity than that between YspE1 and *Shigella* IpaH 9.8. Of note, the sequence differences mainly reside in the LRR domain rather than the NEL domain (Fig. S4A). To find out which part of the YspE1/YspE2 homologs in IP2666pIB1 (DN756_03855 and DN756_03865, respectively, according to the genome annotation of IP2666pIB1) contributes to its divergent capability in hGBPs degradation, we performed domain-swapping experiments between YspE1 and DN756_03855 and between YspE2 and DN756_03865. The results demonstrated that the NEL domains of DN756_03855 and DN756_03865 were catalytic active, as the hybrid protein LRR_{YspE1}-NEL_{IP2666pIB1} and LRR_{YspE2}-NEL_{IP2666pIB1}, but not LRR_{IP2666pIB1}-NEL_{YspE1} and LRR_{IP2666pIB1}-NEL_{YspE2} degraded hGBP1, 2, 5 and 6 (Figure 4, Fig. S4). These results demonstrated that sequence differences in LRR domain of IP2666pIB1 YspE1/YspE2 homologs lead to their inability to degrade hGBPs.

We also included YspE1/YspE2 homologs in YpIII and PB1+ strains, in addition to those in IP2666pIB1, to perform domain-swapping experiment, since they can serve as the representatives of

the three major clusters of YspE1/YspE2 homologs in *Y. pseudotuberculosis* according to the sequence alignment (Fig. S4).

Besides, although both IpaH9.8 and YspE1 target GBPs, they have different substrate preferences to hGBPs and mGBPs (IpaH9.8 degrades hGBP1, 2, 3 and 6, and mGBP2, 3, 6, 7, 9, 10, and 11, whereas YspE1 degrades hGBP1, 2, 5, 6 and mGBP1, 3, 6, 7,10, 11. Li, 2017, Nature). We suppose that some crucial structural confirmation for the binding of YspE1 to hGBPs and mGBPs might be absent in the IP2666pIB1 homolog; however, the detailed molecular mechanism needs to be further determined. We have added the newly acquired data as well as the discussions in the revised manuscript. Please refer to the lines 166-176 and 379-387.

3. Lines 220-237; Fig. 5: I strongly recommend removing the YPIII data from this section for the same reasons as stated above.

Response: It has been removed according to your suggestion.

4. Lines 220-237, Fig. 5: I am still concerned with the statement that you are analyzing intracellular bacteria in Fig. 5. Based on the description in the methods (lines 505-517), one would still predict the majority of the *Y. pestis* should be extracellular in this assay – methods state that bacteria were grown at 26 then shifted to 37 for 2 h prior to infection (this would induce the T3SS and inhibit phagocytosis, especially during infection of Hela cells that are not professional phagocytes). The exception might be YPIII, which might still be expressing Inv to promote uptake. What percentage of these bacteria are taken up by Hela cells (these are not professional phagocytes) and what percentage are simply adhered to the cells? I assume the figures are compressed Z stacks (though not mentioned in the legend). If these were intracellular bacteria in a phagosome, slices should show a ring of GBP around the bacteria. As previously mentioned by another reviewer, higher magnification/resolution images would help demonstrate this (images should include scale bars). Also, doing an ANOVA on Pearson's Correlation Coefficient is not appropriate. The Pearson's Correlation Coefficient measures the strength and direction of a linear relationship between two variables. For colocalization studies, the values range between 0 and 1, with 1 indicating a direct correlation (more likely to colocalize) and 0 indicating no correlation (more likely to not colocalize). By adding the ANOVA, you are doing a statistical analysis on a statistical analysis. Reporting the Pearson's Correlation Coefficients is sufficient.

Response: Thanks for your helpful suggestions. For the better observation of the internalized *Y. pestis* bacilli, we conducted this experiment using bacteria grown at 26 °C without transferring to 37 °C prior

to infection and thus the bacteria are ready to be taken up by the cells. In addition, we have included the bright-field image and multiple slice images to help determine the co-localization signals of hGBP1 with the bacteria residing inside the cells. It is regretfully that we have not obtained the high quality images clearly showing a ring of hGBP1 around the bacteria, probably due to the resolution of the instrument we used is still not high enough. In Fig 6A, the left 4 columns showed only one slice image and the far right column showed the 3D reconstruction images from the multiple slice images of the enlarged framed areas. It was clearly showed that the internalized $\Delta yspE2$ bacteria were largely co-localized with hGBP1 and much less co-localization was observed for the wild-type *Y. pestis* bacteria. Please refer to the Figure 6A in the revised manuscript.

5. Line 295: “Hypersusceptible” seems to be an over statement of the data. There is only a 24 h difference when all animals reach end point and the mean time death appears to be equal for both strains of mice infected with 201. There is likely no difference in LD50 between the mouse backgrounds. It seems that “more susceptible” is a better description of the phenotype. Moreover, because this phenotype is so subtle, it seems that the data from the second independent experiment should be included/reported to ensure that this is a reproducible phenotype. I agree that there might be differences in susceptibility between the 3418 mutant, but this data does not convince me that these mice are hypersusceptible to WT *Y. pestis* infection.

Response: Thank you for pointing out our over statement of the data. We agree that the difference between the mouse backgrounds will not likely be significant according to the survival curves,

although we have not determined the LD₅₀ of 201 strain in *Gbp*^{chr3^{-/-}, chr5^{-/-}} mice. We have modified the description to be “more susceptible” according to your suggestion. Besides, we might have not clearly presented that actually all the experiments for survival curves have been performed for 3 independent times and yielded similar results (please refer to the legend for Figure 8), and only one representative result was shown. The results for the rest replicates can be found in the raw data of this manuscript.

6. Line 351-353: For reasons stated above, I do not believe the authors can conclude that they have demonstrated that the proteins from Pa3606 can degrade the GBPs. The case of YPIII is even weaker, as all of the data from this strain was performed with a strain that lacks the ability to secrete the proteins into host cells.

Response: Thanks for your comments. I guess you made a typo in the first sentence and actually did not believe that proteins from Pa3606 or YPIII cannot degrade the GBPs. Pa3606 contains the pYV plasmid that encodes a T3SS capable of translocating proteins. However, YspE1/YspE2 homologs in Pa3606 contain only LRR domain but no intact NEL domain according to the sequence alignment. Thus, it is reasonable that Pa3603 infection fails in degradation of GBPs. For YPIII, by using cell transfection assay, we demonstrated that both of the YspE1/YspE2 homologs in YPIII cannot degrade any of hGBPs. With these additional data, we can conclude that YspE1/YspE2 homologs from Pa3606 and YPIII cannot degrade hGBPs. Please refer to Figure 4 and Fig. S4 in the revised manuscript.

7. Line 369-370: The data do not definitively show that the Y *pstb* homologs do not degrade the GBPs.

Response: We have cloned the YspE1/YspE2 homologs in IP2666I and YPIII, as well as the YspE1 homolog in PB1+ into 3 × FLAG-pCMV vector, and the successful expressions of these proteins were confirmed by immunoblotting analysis. HeLa cells expressing hGBP1-7 were individually transfected with these plasmids and the levels of hGBPs were analyzed. The results demonstrated that the YspE1/YspE2 homologs in IP2666I and YPIII, as well as the YspE1 homolog in PB1+, cannot degrade any of the hGBPs. Please refer to Figure 4 and Fig. S4 in the revised manuscript.

Additional Minor comments:

1. The switching of between *yp-3416/3418* and *yspE1/E2* has become more confusing in the new draft (even individual figures use both designations). Suggest that you standardize nomenclature throughout the manuscript

Response: We are awfully sorry for the confusion caused by the protein/gene designations. In the previous version of manuscript, we use *yp-3416/3418* when refer to genes and use YspE1/YspE2 when

refer to proteins, but it seems to be not as clear as we thought. According to your suggestions, we have standardized nomenclature throughout the manuscript.

2. Lines 274-281, Fig. 6J: I had to reread this section a couple times before it was clear that you ultimately concluding that you could not detect differences in inflammasome activation in macrophages from chr 3/5 mice. During my first read, it appeared that you were over interpreting your data in Fig 6J with the conclusion that inflammasome activation is even occurring in these cells. Based on the representative image for 6J, Casp 1 expression is extremely low in macrophages from chr 3/5 mice (see mock treated cells), and it appears that a much longer exposure was required to see any bands in infected cells (based on the greater appearance of secondary bands in cell lysate and supernatant). Perhaps it would be better to simplify this statement. For example, “Due to low levels of Casp 1 in macrophages from Chr 3/5 mice, we were unable to detect significant inflammasome activation in these cells during *Y. pestis* infection (Fig. 6J). I don’t think these cells are required to make the overall conclusion of this section.

Response: We highly appreciate your comments and suggestions, and admit that our original statements on inflammasome activation in *Gbp^{chr3-/-,chr5-/-}* BMDM cells were obscure and hard to understand. These statements have been modified to be more succinct and precise according to your suggestions in the revised manuscript. Please refer to the lines 277-280.

3. Fig 6: This is the first figure that you designate *Y. pestis* 201 as WT. All previous figures use the 201 designation. Recommend that you use consistent nomenclature throughout to reduce confusion by readers.

Response: Thanks for your kind reminding. The inconsistency of *Y. pestis* 201 designation have been corrected in the revised manuscript.

4. Fig. 6: How long were macrophages incubated with 30 ug/ml gent. before changing to 2 ug/ml and how does this relate to the time points in figure 6 (does 2 h post infection reflect the time at which the 30 ug was removed)? This should be clear for others who might want to reproduce your data, but is also important for the reader to figure out total bacteria added vs. associated vs. intracellular.

Response: We are sorry for the confused description of the gentamicin protection assay. The infected macrophages were incubated with DMEM containing 10% FBS and 30 µg/ml gentamicin at 0.5 hpi. to kill the extracellular bacteria. After incubation for additional 0.5 h, the culture medium was replaced with fresh DMEM containing 10% FBS and 2 µg/ml gentamicin, which was then sustained till the end

of the experiment. The time point when the bacterial suspensions were added into the cells were designed as the zero point. Thus, it was 1 hour post infection when the culture medium containing 30 µg/ml gentamicin was removed. We have improved the description of this experiment and hope it will be clear enough. Please refer to lines 496-499.

5. Lines 289-291; Fig. 7: Include a description of when these tissues were harvested and bacterial burdens were determined. Fig. 7B and D require inclusion of the limit of detection for each tissue. Authors may also want to consider changing this to a box and whisker graph with all points, as it might better represent the differences.

Response: Thanks for your helpful advices. We are sorry for having not described the time point clearly for the determination of bacterial burdens. We have supplemented both the text and the legend of this figure with the following experimental details. The tissues were harvested when the mice were moribund, i.e. 4 day post infection (dpi.) for C57BL/6J mice and 3 dpi. for *Gbp*^{chr3-/-,chr5-/-} mice, due to the more susceptibility of *Gbp*^{chr3-/-,chr5-/-} mice to *Y. pestis*. The detection limits for each tissue were calculated according to the raw data and were as follows: 21 CFU/100 mg lung tissues, 40 CFU/spleen, 80 CFU/groin lymph node, 19 CFU/100 mg liver tissues. We have modified the graphs of this figure according to your suggestions in the revised manuscript and the limit of detection were supplemented in the legend of this figure. Please refer to Figure 8 in the revised manuscript.

6. Fig. 7B. It seems that a one way ANOVA is the more appropriate analysis (there is only one variable, the strain used to infect the animals).

Response: Thanks for pointing out our mistake. It has been corrected and the statistical analysis of this figure was performed using one-way ANOVA. Please refer to Figure 8B in the revised manuscript.

7. Fig. 7D: This data would be easier for the reader to interpret if it was grouped by tissue not mouse strain (similar to 7B).

Response: Thanks for your advice. The data has been regrouped by tissue and it seems to be much clearer now. Please refer to Figure 8D in the revised manuscript.

8. Line 263-266: It is not stated anywhere how long the infections took place before they were measured for inflammasome activation. Would later time points show activation?

Response: Thanks for your comments. We are sorry for having not explained this point clearly. Samples for inflammasome activation assay were collected at 4 hpi. as that has been described in the

methods section. For clarity, these experimental details have been provided in the text (please refer to the line 269). We have analyzed the inflammasome activation in cells after infection for as long as 8 hours, and no activation of inflammasome has been found.

Reviewer #2 (Remarks to the Author):

The authors have made reasonable efforts to address my questions.

However, questions remain:

1.) The ability of NEL ligases (specifically IpaH9.8) to degrade GBPs and to antagonize host immunity is well established (Li Nature 2017, Wandel Cell Host Microbe 2017, Wandel Nat Immunol 2020). Despite pointing out these points in my original summary of their paper, the authors keep avoiding the issue. In my opinion, these points need to be addressed straightforwardly in the introduction and should not be hidden somewhere in the text.

Response: Thanks for your comments. We are sorry that you feel we are keep avoiding to point out the well-established ability of NEL ligases (specifically IpaH9.8) to degrade GBP. Actually, we have introduced these findings in the first part of Results when we are talking about the sequence analysis of *yspE1* and *yspE2*, revealing that they are homologous to IpaH9.8. For clarity, a description of NEL ligases (specifically IpaH9.8) activity to degrade GBPs and to antagonize host immunity have been added in the Introduction section. Please refer to the lines 47-50 in the revised manuscript.

2.) Former Major Point 1: It is regrettable that the authors failed to quantify the extent to which *Yersinia* invades the host cytosol. Galectins are well established tools and work without problems in HeLa cells. Have they tried any positive controls (*Listeria*, *Shigella*, *Salmonella*) to test their galectin constructs and did they try to stain endogenous galectins if their expression system does not work? Regarding the colocalization of GBPs and *Yersinia* – is the Pearson's coefficient now reported in Fig5F really informing us about the percentage of bacteria associated with GBPs as claimed by the authors in their rebuttal?

Response: Thanks for your helpful suggestions. We have optimized the experimental conditions and managed to detect the *Y. pestis* bacilli that invade the cytosol. IFN- γ primed HeLa cells were infected with various RFP-expressing *Y. pestis* strains and hGBP1 and galectin-8 were stained at 1 hour post infection. The results showed that hGBP1 significantly co-localized with Δ *yspE2* but rarely with the

wild-type *Y. pestis* 201, due to its degradation by YspE1/2, indicating that the membrane of $\Delta yspE2$ -containing vacuoles was damaged and galectin-8 was recruited subsequently. It has been shown that significantly fewer bacteria were targeted by galectin-8 in *Gbp^{chr3-/-}* BMDMs than in wild-type macrophages, suggesting that GBPs might promote the lysis of vacuoles or help to recruit galectin-8 to the lysed vacuoles (Meunier E, et al, 2014). Our results are in line with this finding that hGBP1 plays role in the lysis of Yersinia-containing vacuoles, and degradation of hGBPs, especially hGBP1, blocks disruption of the vacuolar membrane and galectin-8 recruitment that initiate the uptake of bacteria into autophagosomes. This part of results has not been added into the revised manuscript because we need to do more investigation to confirm our present findings.

3.) The title is still non-senical.

Response: Thanks for your comments. We have modified the title to be “*Yersinia pestis* E3 ligases ubiquitinate and degrade GBPs to antagonize the host cell-autonomous defense —— a novel arsenal acquired during evolution”, and hope it will meet your requirement.

Reviewer #3 (Remarks to the Author):

Authors have made a clear effort to answer all the points raised and the manuscript is now acceptable for publication.

References

1. Meunier E, Dick MS, Dreier RF, Schürmann N, Kenzelmann Broz D, Warming S, Roose-Girma M, Bumann D, Kayagaki N, Takeda K, Yamamoto M, Broz P. Caspase-11 activation requires lysis of pathogen-containing vacuoles by IFN-induced GTPases. *Nature*. 2014 May 15;509(7500):366-70. doi: 10.1038/nature13157. Epub 2014 Apr 16. PMID: 24739961.

REVIEWERS' COMMENTS

Reviewer #1 (Remarks to the Author):

The careful considerations made by the authors to address the previous comments have resulted in a much better manuscript. I believe the authors have addressed all of my previous concerns.

Reviewer #2 (Remarks to the Author):

The authors have made further attempts to address my comments but the manuscript still fails to provide information on the fraction of Yersinia that enter the cytosol and become coated with GBPs. I find it surprising that such a simple question is not addressed after several rounds of review. (Pearson's correlation coefficient does not provide this information (page 12), in contrast to the authors' claim in their last rebuttal; this time my question was simply left unanswered.)

Furthermore, on the topic of providing a fair representation of previous knowledge, I do not understand why the authors are not willing to make clear statements. They write: "... Bacterial pathogens including Shigella, Francisella, Legionella and Salmonella have evolved multiple virulence effectors to disrupt GBP-mediated host defense²⁷⁻²⁹. For instance, Shigella IpaH9.8, a T3SS effector, can target hGBPs for degradation to promote the intracellular replication and spread of bacteria^{30,31}. In this study, we showed that two novel E3 ubiquitin ligase (NEL) family members, YP_3416 and YP_3418, which were named as Yersinia secreted E3 ligase 1 (YspE1) and YspE2 in this study, ubiquitinate GBPs for proteasomal degradation of both human GBPs (hGBPs) and murine GBPs (mGBPs), to promote the survival inside the macrophages and strongly inhibit the inflammasome activation."

The non-specialist reader could be forgiven to believe the authors have discovered that NEL family members ubiquitylate GBPs and target them for proteasomal degradation. In reality, all this was known from Li *Nature* 2017 and Wandel *Cell Host Microbes* 2017, who demonstrated that the Shigella NEL ligase IpaH9.8 does exactly what Cao et al. claim they have discovered – the NEL ligase IpaH9.8 ubiquitylates and targets GBPs for proteasomal degradation. What Cao et al. reveal here is that two Yersinia homologues of IpaH9.8 do the same. Is that worth reporting? Yes, it certainly is of interest to the specialist reader who cares about Yersinia. But even the Yersinia specialist would appreciate to read that the phenomenon itself has been known for many years.

Response to Referees

Reviewers' comments:

Reviewer #1 (Remarks to the Author):

The careful considerations made by the authors to address the previous comments have resulted in a much better manuscript. I believe the authors have addressed all of my previous concerns.

Reviewer #2 (Remarks to the Author):

The authors have made further attempts to address my comments but the manuscript still fails to provide information on the fraction of *Yersinia* that enter the cytosol and become coated with GBPs. I find it surprising that such a simple question is not addressed after several rounds of review. (Pearson's correlation coefficient does not provide this information (page 12), in contrast to the authors' claim in their last rebuttal; this time my question was simply left unanswered.)

Response: Thanks to your previous suggestions, we have managed to detect the *Y. pestis* bacilli that invade the cytosol using galectin-8 as a marker by optimizing the experimental conditions, but it is difficult to distinguish the individual bacterium for accurately counting the number of bacteria colocalized with galectin-8, probably due to the resolution of the instrument we used is not high enough. This is why we have not included these results in the main manuscript and we believe that we need to further improve our experimental settings to reach an accurate conclusion on this issue. Based on the current data, we can only evaluate approximately the fraction of *Y. pestis* that can enter the cytosol. IFN- γ primed HeLa cells were infected with various RFP-expressing *Y. pestis* strains and hGBP1 and galectin-8 were stained at 1 hour post infection. It seems that 80~90% of $\Delta yspE2$ bacteria were colocalized with both hGBP1 and galectin-8, for the wild-type *Y. pestis* bacteria the fraction were less than 20%. We speculate that degradation of GBPs, especially hGBP1, blocks the disruption of the vacuolar membrane and galectin-8 recruitment that can initiate the uptake of *Y. pestis* bacilli into autophagosomes. This is an interesting issue worthy for further investigation.

Furthermore, on the topic of providing a fair representation of previous knowledge, I do not

understand why the authors are not willing to make clear statements. They write: "... Bacterial pathogens including *Shigella*, *Francisella*, *Legionella* and *Salmonella* have evolved multiple virulence effectors to disrupt GBP-mediated host defense²⁷⁻²⁹. For instance, *Shigella* IpaH9.8, a T3SS effector, can target hGBPs for degradation to promote the intracellular replication and spread of bacteria ^{30,31}. In this study, we showed that two Novel E3 ubiquitin ligase (NEL) family members, YP_3416 and YP_3418, which were named as *Yersinia* secreted E3 ligase 1 (YspE1) and YspE2 in this study, ubiquitinate GBPs for proteasomal degradation of both human GBPs (hGBPs) and murine GBPs (mGBPs), to promote the survival inside the macrophages and strongly inhibit the inflammasome activation."

The non-specialist reader could be forgiven to believe the authors have discovered that NEL family members ubiquitylate GBPs and target them for proteasomal degradation. In reality, all this was known from Li Nature 2017 and Wandel Cell Host Microbes 2017, who demonstrated that the *Shigella* NEL ligase IpaH9.8 does exactly what Cao et al. claim they have discovered – the NEL ligase IpaH9.8 ubiquitylates and targets GBPs for proteasomal degradation. What Cao et al reveal here is that two *Yersinia* homologues of IpaH9.8 do the same. Is that worth reporting? Yes, it certainly is of interest to the specialist reader who cares about *Yersinia*. But even the *Yersinia* specialist would appreciate to read that the phenomenon itself has been known for many years.

Response: Thanks for your comments. We are sorry that you feel that we are not willing to make clear statements. We do not think that non-specialist reader could believe that we have discovered that NEL family members ubiquitylate GBPs and target them for proteasomal degradation. In addition to the description in the Introduction section, which states IpaH9.8 activity to degrade GBPs and to antagonize host immunity, we also clearly stated in the first part of Results that "It has been reported that *Shigella* IpaH9.8 targets hGBPs for ubiquitination and proteasomal degradation, which ultimately promotes the intracellular replication, motility and spread of *S. flexneri* bacteria." Even though, for addressing your concerns, the description has been modified to be as follows and we hope it will fulfill your requirement:

".....Bacterial pathogens including *Shigella*, *Francisella*, *Legionella* and *Salmonella* have evolved multiple virulence effectors to disrupt GBP-mediated host defense. For instance, *Shigella* IpaH9.8, a T3SS effector, can ubiquitinate hGBPs for proteasomal degradation to promote the intracellular replication and spread of bacteria. In this study, we showed that two Novel E3 ubiquitin ligase (NEL) family members, YP_3416 and YP_3418, which were named as *Yersinia* secreted E3 ligase 1 (YspE1) and YspE2 in this study, target GBPs for degradation of both human GBPs (hGBPs) and

murine GBPs (mGBPs), to promote the survival inside the macrophages and strongly inhibit the inflammasome activation.”